# Graph-GRPO: Training Graph Flow Models with Reinforcement Learning

**Baoheng Zhu** [* 1]   **Deyu Bo** [* 2]   **Delvin Ce Zhang** [3]   **Xiao Wang** [† 4]

## Abstract

Graph generation is a fundamental task with broad applications, such as drug discovery. Recently, discrete flow matching-based graph generation, *a.k.a.*, graph flow model (GFM), has emerged due to its superior performance and flexible sampling. However, effectively aligning GFMs with complex human preferences or task-specific objectives remains a significant challenge. In this paper, we propose Graph-GRPO, an online reinforcement learning (RL) framework for training GFMs under verifiable rewards. Our method makes two key contributions: (1) We derive an analytical expression for the transition probability of GFMs, replacing the Monte Carlo sampling and enabling fully differentiable rollouts for RL training; (2) We propose a refinement strategy that randomly perturbs specific nodes and edges in a graph, and regenerates them, allowing for localized exploration and self-improvement of generation quality. Extensive experiments on both synthetic and real datasets demonstrate the effectiveness of Graph-GRPO. With only 50 denoising steps, our method achieves 95.0% and 97.5% Valid-Unique-Novelty scores on the planar and tree datasets, respectively. Moreover, Graph-GRPO achieves state-of-the-art performance on the molecular optimization tasks, outperforming graph-based and fragment-based RL methods as well as classic genetic algorithms. Code is available in https://github.com/Zhubaoheng/Graph-GRPO.

## 1. Introduction

Recent advances in discrete generative modeling (Austin et al., 2021; Gat et al., 2024) have given rise to a wide range of graph generation methods (Vignac et al., 2023; Xu

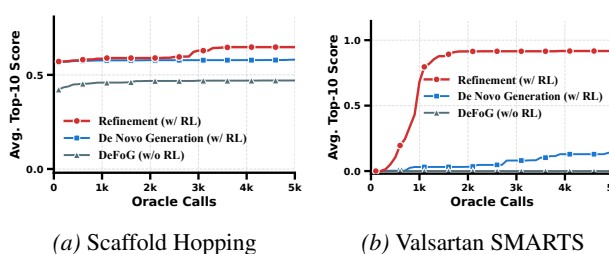

*(a)* Scaffold Hopping     *(b)* Valsartan SMARTS

*Figure 1.* Reward curves on two molecular optimization tasks. We use DeFoG (Qin et al., 2025) as the base model. As the number of oracle calls increases, the score of RL-optimized models gradually rises, while the base model remains almost unchanged. In the tasks with highly selective reward, *e.g.*, Valsartan SMARTS, refining promising candidates is more effective than de novo generation.

et al., 2024; Siraudin et al., 2025). Among them, discrete flow matching-based graph generation methods, commonly referred to as graph flow models (GFMs), have recently attracted increasing attention. By decoupling the training objective from sampling, GFMs enable more effective generative modeling and flexible inference (Qin et al., 2025).

Despite significant progress, existing GFMs remain limited in satisfying complex human preferences or task-specific objectives. For example, drug discovery requires generating small molecules with high binding affinity and low toxicity (Xu et al., 2023). However, generating such molecules remains computationally expensive due to the vast space of generative models. Recently, online reinforcement learning (RL) (Schulman et al., 2017) has demonstrated significant promise in aligning generative models with human priors by maximizing specific reward functions. While existing studies have explored the integration of RL with graph generative adversarial networks (You et al., 2018a) and graph diffusion models (Liu et al., 2024), extending RL to GFMs introduces two fundamental challenges:

- Modern RL algorithms rely on policy gradients (Sutton et al., 1999) for optimization, which requires the policy model to be differentiable for the transition probability of each action. However, existing GFMs estimate the action probabilities via Monte Carlo sampling (Bishop & Nasrabadi, 2006), breaking the gradient flows.

- RL exploration needs sufficient feedback to locate the task-specific region in the generative space. However, GFMs typically perform de novo generation, which can

---

[*]Equal contribution [1]Beijing University of Posts and Telecommunications [2]National University of Singapore [3]University of Sheffield [4]Beihang University. Correspondence to: Xiao Wang <xiao_wang@buaa.edu.cn>.

*Proceedings of the 43rd International Conference on Machine Learning*, Seoul, South Korea. PMLR 306, 2026. Copyright 2026 by the author(s).

produce sparse reward signals, *i.e.*, most generated graphs are invalid, leading to ineffective explorations.

To address these two challenges, we propose Graph-GRPO, an online RL framework that uses Group Relative Policy Optimization (GRPO) (Shao et al., 2024) to align GFMs with task-specific objectives. Specifically, we first derive an analytical expression of GFMs that explicitly connects model predictions to the action probability, which enables fully differentiable sampling and allows GFMs to be trained with modern RL frameworks. Furthermore, we propose a refinement strategy to efficiently explore the potentials of promising samples. For graphs with higher reward scores, we iteratively inject a small amount of noise and let GFMs regenerate clean graphs. By repeating this process, Graph-GRPO progressively concentrates on high-potential regions of the generative space and improves generation quality over multiple refinement rounds. Figure 1 illustrates the effectiveness of the proposed refinement strategy on two molecular optimization tasks. While de novo generation and refinement achieve comparable performance on the basic Scaffold Hopping task, a large performance gap emerges on the more challenging Valsartan SMARTS task, demonstrating the necessity and effectiveness of iterative refinement in complex optimization scenarios. The contributions of this paper are summarized below:

- We propose Graph-GRPO, an online RL framework that enables end-to-end RL training of GFMs by replacing the non-differentiable Monte Carlo sampling with an analytical transition probability.

- We introduce an iterative refinement strategy that refines high-reward samples through controlled perturbation and regeneration, enabling localized exploration of promising regions in the chemical space.

- Extensive experiments on synthetic graph benchmarks and molecular design tasks demonstrate the state-of-the-art performance of Graph-GRPO, outperforming existing RL-based and evolutionary approaches.

## 2. Preliminary

**Discrete State Graph.** Let $G = \{x^{1:N}, e^{1:N^2}\}$ denote an undirected graph with $N$ nodes and $N^2$ edges, where $x^i \in \{1, \cdots, \mathcal{X}\}$ and $e^j \in \{1, \cdots, \mathcal{E}\}$ denote the discrete categories of nodes and edges, respectively.

Each graph is represented as $N + N^2$ dimensional data, and has $|\mathcal{X}|^N \times |\mathcal{E}|^{N^2}$ possibilities. It is intractable to model the transition probability between graphs, and we address this issue by factorizing a graph into independent nodes and edges (Campbell et al., 2022):

$$p(G) = \prod_{i=1}^{N} p(x^i) \prod_{j=1}^{N^2} p(e^j). \quad (1)$$

For brevity, we omit the index of nodes and edges and use a uniform notation $z$ to represent $x$ and $e$, where $\mathcal{Z} = \mathcal{X} \cup \mathcal{E}$.

### 2.1. Discrete Flow Matching

We consider a time-indexed stochastic process $\{z_t\}_{t \in [0,1]}$ defined on $\mathcal{Z}$, where $p_t(z = z_t)$ denotes the marginal distribution of $z_t$ at time $t$. Discrete flow matching aims to construct a continuous probability path $\{p_t\}_{t \in [0,1]}$ connecting a prior distribution $p_0$ to the data distribution $p_1$.

**Noising Process** ($t : 1 \to 0$). In the noising process, given a data state $z_1$, the conditional probability path is defined via linear interpolation:

$$p_{t|1}(z_t \mid z_1) = t \cdot \delta(z_t, z_1) + (1 - t) \cdot p_0(z_t), \quad (2)$$

where $\delta(i, j)$ denotes the Kronecker delta function, *i.e.*, $\delta(i, j) = 1$ if $i = j$ and 0 otherwise.

**Denoising Process** ($t : 0 \to 1$). The denoising process is modeled as a Continuous-Time Markov Chain (CTMC) that evolves the distribution from prior $p_0$ to data $p_1$. The transition probability from $z_t$ to $z_{t+dt}$ is defined as:

$$p(z_{t+dt}|z_t) = \begin{cases} R_t(z_t, z_{t+dt})dt & \text{if } z_{t+dt} \neq z_t, \\ 1 + R_t(z_t, z_t)dt & \text{if } z_{t+dt} = z_t, \end{cases} \quad (3)$$

where $R_t$ is the rate matrix that determines whether to rest in the current state or jump to another state, and $R_t(z_t, z_t) = -\sum_{z_{t+dt} \neq z_t} R_t(z_t, z_{t+dt})$. In the following, we denote $p(z_{t+dt}|z_t)$ as the transition probability of the specific state, and use $p(\cdot|z_t) = [p(1|z_t), \cdots, p(|\mathcal{Z}| \mid z_t)]$ to represent the distribution of all states.

### 2.2. Training and Sampling

**Continuous-time Training.** GFMs aim to learn a graph denoiser $f_\theta$ that takes the graphs with different noise levels as input and predicts the clean graphs, formulated as:

$$\mathcal{L} = \mathbb{E}_{t \sim (0,1), G_1 \sim p_1, G_t \sim p_{t|1}} \text{CE}\big(G_1, p_\theta(\cdot|G_t)\big), \quad (4)$$

where CE denotes the cross-entropy loss and $p_\theta(\cdot|G_t)$ is the probability distribution predicted by the denoiser $f_\theta(G_t, t)$.

**Discrete-time Sampling.** The sampling process generates data from noise by simulating the trajectory of a CTMC using a finite time interval $\Delta t$. Given the current discrete state $z_t$, the next state $z_{t+\Delta t}$ is sampled from the categorical distribution defined by the transition probabilities:

$$z_{t+\Delta t} \sim \text{Categorical}\big(p_{t+\Delta t|t}(\cdot|z_t)\big), \quad (5)$$

where the rate matrix is determined by the denoising model prediction $p_\theta$ and the initial prior distribution $p_0$. This process is repeated iteratively from $t = 0$ to 1.

---

**Algorithm 1** Conditional Transition (DeFoG)

**Input:** Graph $G_t$, time $t$, interval $\Delta t$, denoiser $f_\theta$

$p_\theta(\cdot|z_t) \leftarrow \text{Softmax}(f_\theta(G_t, t))$ ▷ Model Prediction

$\hat{z}_1 \leftarrow \text{Categorical}(p_\theta(\cdot|z_t))$ ▷ Monte Carlo Sampling
$R_t(z_t, z_{t+\Delta t}) \leftarrow R_t(z_t, z_{t+\Delta t}|\hat{z}_1)$ ▷ Eq. 8

$p(z_{t+\Delta t}|z_t) = \delta(z_t, z_{t+\Delta t}) + R_t(z_t, z_{t+\Delta t})\Delta t$ ▷ Transition

**Output:** $p_{t+\Delta t} = [p(1|z_t), \cdots, p(|\mathcal{Z}| \mid z_t)]$

---

**Algorithm 2** Analytic Transition (Graph-GRPO)

**Input:** Graph $G_t$, time $t$, interval $\Delta t$, denoiser $f_\theta$

$p_\theta(\cdot|z_t) \leftarrow \text{Softmax}(f_\theta(G_t, t))$ ▷ Model Prediction

$R_t^\theta(z_t, z_{t+\Delta t}) \leftarrow p_\theta(z_{t+\Delta t})V_1 + (1 - p_\theta(z_t) - p_\theta(z_{t+\Delta t}))V_2$
▷ Analytic Rate Matrix, Eq 10

$p(z_{t+\Delta t}|z_t) = \delta(z_t, z_{t+\Delta t}) + R_t^\theta(z_t, z_{t+\Delta t})\Delta t$ ▷ Transition

**Output:** $p_{t+\Delta t} = [p(1|z_t), \cdots, p(|\mathcal{Z}| \mid z_t)]$

---

## 3. The Proposed Method

This section details the proposed method. We begin by deriving the analytical transition probability for GFMs, and then introduce RL training and refinement in Graph-GRPO.

### 3.1. Estimation of Rate Matrix

In each denoising step of GFMs, we need to estimate a rate matrix $R_t$ to transform the graph from $G_t$ to $G_{t+dt}$. The marginal distribution $p_t$ and the rate matrix $R_t$ are connected by the Kolmogorov forward equation:

$$\partial_t p_t = R_t^\top p_t. \qquad (6)$$

Satisfying the above condition is challenging because it is difficult to directly obtain the differentiation of $p_t$. On the other hand, if we can access the real data $z_1$, it is possible to give the conditional differentiation:

$$\partial_t p_{t|1}(z_t \mid z_1) = \delta(z_t, z_1) - p_0(z_t). \qquad (7)$$

**Conditional Rate Matrix.** Given a clean state $z_1$, Campbell et al. (2024) gives a valid conditional rate matrix:

$$R_t(z_t, z_{t+dt}|z_1) = \frac{\text{ReLU}\big[\partial_t p_{t|1}(z_{t+dt}|z_1) - \partial_t p_{t|1}(z_t|z_1)\big]}{Z_t^{>0} \cdot p_{t|1}(z_t|z_1)}, \qquad (8)$$

where $Z_t^{>0} = \big|\{z_t : p_{t|1}(z_t|z_1) > 0\}\big|$ denotes the number of states with non-zero probability.

Since the ground-truth state $z_1$ is unknown during generation, existing GFMs employ a Monte Carlo sampling to sample a pseudo-graph as an alternative to the real data. As the number of samples increases, the conditional rate matrix gradually transforms into the unconditional rate matrix:

$$R_t(z_t, z_{t+dt}) = \mathbb{E}_{\hat{z}_1 \sim p_\theta(\cdot|z_t)} [R_t(z_t, z_{t+dt}|\hat{z}_1)], \qquad (9)$$

where $\hat{z}_1$ denotes the pseudo-state sampled from $p_\theta(\cdot|z_t)$. The transition process of GFMs is shown in Algorithm 1. Although effective, the conditional rate matrix is not suitable for RL training for two reasons:

- **Non-differentiable transition probability.** The RL optimization relies on calculating a ratio between the old and new policy models, $r = \frac{\pi_\theta(G_{t+dt}|G_t)}{\pi_{\text{old}}(G_{t+dt}|G_t)}$, which requires the policy model $\pi$ to be fully differentiable. However, the conditional rate matrix relies on non-differentiable Monte Carlo sampling, which breaks the gradient flow.

- **Mismatch between training and inference.** Even though we can use some tricks, *e.g.*, Gumbel-Softmax, to keep the gradient continuous, there is a more serious mismatch issue in the conditional rate matrix. That is, the new and old models may sample different pseudo-graphs, resulting in inconsistency between training and inference (Ma et al., 2025; Li et al., 2026; Li & Wang, 2026a;b).

**Analytical Rate Matrix.** To enable stable RL training, we theoretically derive the analytical expression of the rate matrix, which is fully differentiable and aligns the training and inference processes.

**Proposition 3.1.** *Given the current state $z_t$, current time $t$, prior distribution $p_0$, and model prediction $p_\theta(\cdot|z_t)$, the off-diagonal entry of the analytic rate matrix is defined as:*

$$R_t^\theta(z_t, z_{t+dt}) = p_\theta(z_{t+dt})V_1 + (1 - p_\theta(z_t) - p_\theta(z_{t+dt}))V_2, \qquad (10)$$

$$V_1 = \frac{1 + p_0(z_t) - p_0(z_{t+dt})}{Z_t^{>0}(1 - t)p_0(z_t)}, \; V_2 = \frac{\text{ReLU}(p_0(z_t) - p_0(z_{t+dt}))}{Z_t^{>0}(1 - t)p_0(z_t)},$$

*where $V_1$ and $V_2$ are two statistics that can be pre-calculated before generation.*

*Proof.* See Appendix A. □

Proposition 3.1 states that the rate matrix can be directly estimated from the predictions of the denoiser by traversing all possibilities of the real data. As a result, the denoiser can be optimized by the policy gradient, and the reward can be maximized. The transition process of Graph-GRPO is shown in Algorithm 2.

### 3.2. Graph-GRPO

In this section, we introduce the pipeline of Graph-GRPO. The framework is shown in Figure 2.

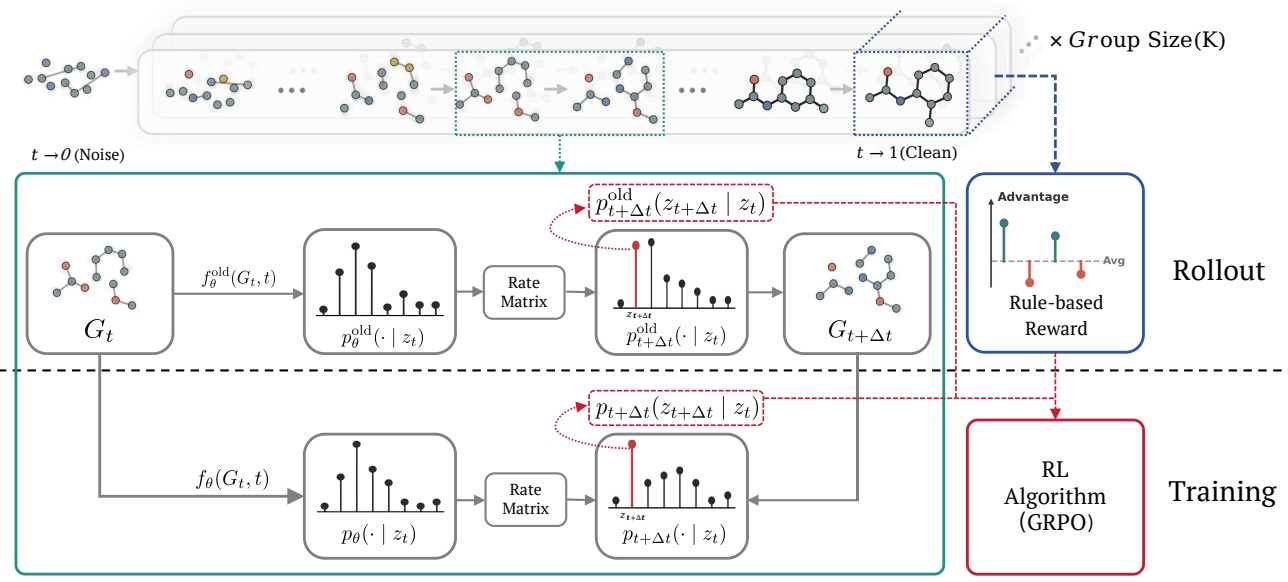

*Figure 2.* The overall framework of Graph-GRPO. (1) **Rollout**: given a noisy graph, the policy model samples $K$ denoising trajectories and caches the graph state $z_{t+\Delta t}$ along with its transition probability $p_{t+\Delta t}^{\text{old}}$. In the meantime, we normalize the reward scores to calculate the advantage of each trajectory. (2) **RL training**: we select a graph $G_t$ from the rollouts and use the new policy model to estimate the transition probabilities $p_{t+\Delta t}$. Graph-GRPO maximizes the rewards by optimizing the advantage-weighted ratio in Eq. 14.

**Rollout Collection.** A rollout refers to a trajectory $\tau$ generated by sequentially sampling actions from the policy model $\pi_\theta$ (Shao et al., 2024), which can be formulated as:

$$\tau = \{s_0, a_0, \cdots, s_T, a_T\}, \quad (11)$$

where $s$ denotes state, $a$ represents an action, and $T$ is the length of the trajectory. In our context, the policy model is a pre-trained GFM, the state is the input graph, and the action samples the nodes and edges of the new graph.

To implement group-relative optimization, we first need to collect a set of trajectories starting from the same origin, *i.e.*, a noisy graph $G_0 \sim p_0$. Conditioned on the same noise graph, GFMs will generate a group of independent trajectories $\{\tau^{(k)}\}_{k=1}^K$ in parallel, where $K$ denotes the group size. During this process, we explicitly record the sampled trajectory $\tau^{(k)} = \{G_0, G_{\Delta t}^{(k)}, \cdots, G_{1-\Delta t}^{(k)}, G_1^{(k)}\}$ along with the transition probability $p(G_{t+\Delta t}^{(k)} | G_t^{(k)})$ at each step.

Upon reaching the terminal state $t = 1$, we evaluate the final graph to compute the reward $R^{(k)} = \mathcal{R}(G_1^{(k)})$, where $\mathcal{R}$ denotes the task-specific reward functions, *e.g.*, structural validity or docking scores. Subsequently, we utilize the cached trajectory $\tau^{(k)}$, the sequence of step-wise transition probabilities, and the reward $R^{(k)}$ to compute the advantage and the importance sampling ratio for RL training.

**RL Training.** The goal of RL is to learn an optimal policy that maximizes the expected reward function. Given the pre-collected rollouts, we aim to learn a new policy model $\pi_\theta$ that achieves higher rewards than the old policy model $\pi_{\text{old}}$, which can be formulated as:

$$\mathcal{J}(\theta) = \mathbb{E}_{\tau \sim \pi_\theta}[R(\tau)] = \mathbb{E}_{\tau \sim \pi_{\text{old}}}\left[\frac{p_\theta(\tau)}{p_{\text{old}}(\tau)} R(\tau)\right], \quad (12)$$

where $\frac{p_\theta(\tau)}{p_{\text{old}}(\tau)}$ is the importance sampling that performs a change of measure between rollout distributions.

The objective function of Graph-GRPO is the same as the standard GRPO framework, which is defined as:

$$\mathcal{J}(\theta) = \frac{1}{KT} \sum_{k=1}^K \sum_{t=1}^T \left(\mathcal{L}_t^{(k)}(\theta) - \beta D_{\text{KL}}(\pi_\theta \| \pi_{\text{ref}})\right), \quad (13)$$

where $\mathcal{L}_t^{(i)}(\theta)$ denotes the policy optimization in the $k$-th rollout and $t$-th denoising step, $\beta$ is a hyperparameter, and $D_{\text{KL}}$ is used to prevent the RL-optimized model $\pi_\theta$ from drastically deviating from the base model. The loss of policy optimization is formulated as follows.

$$\mathcal{L}_t^{(k)}(\theta) = \min\left(r_{t,\theta}^{(k)} A^{(k)}, \text{clip}(r_{t,\theta}^{(k)}, 1 \pm \epsilon) A^{(k)}\right), \quad (14)$$

$$A^{(k)} = \frac{R^{(k)} - \text{mean}(\{R^{(i)}\})}{\text{std}(\{R^{(i)}\})}, \quad r_{t,\theta}^{(k)} = \frac{\pi_\theta(G_{t+\Delta t}^{(k)}|G_t^{(k)})}{\pi_{\text{old}}(G_{t+\Delta t}^{(k)}|G_t^{(k)})},$$

where $A^{(k)}$ denotes the group-relative advantage and $r_{t,\theta}^{(k)}$ is the importance sampling ratio. Moreover, to prevent reward hacking and preserve chemical validity, we use the KL divergence to penalize deviations:

$$D_{\text{KL}}(\pi_\theta \| \pi_{\text{ref}}) = \pi_\theta(G_{t+\Delta t}^{(k)}|G_t^{(k)}) \log \frac{\pi_\theta(G_{t+\Delta t}^{(k)}|G_t^{(k)})}{\pi_{\text{ref}}(G_{t+\Delta t}^{(k)}|G_t^{(k)})}, \quad (15)$$

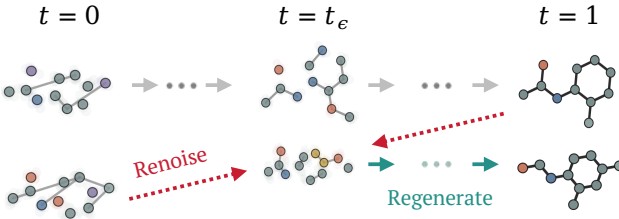

$t = 0 \qquad t = t_\epsilon \qquad t = 1$

Renoise

Regenerate

*Figure 3.* Refinement in Graph-GRPO. We first use GFMs to denoise a noise graph from $t = 0$ to $t = 1$. Subsequently, we re-noise the generated graph to time step $t_\epsilon$ and denoise it again using GFMs. This strategy explicitly increases the denoising steps of GFMs and improves the generation quality of Graph-GRPO.

where $\pi_{\text{ref}}$ is the base model, *e.g.*, a pre-trained DeFoG checkpoint without RL training.

### 3.3. Refinement

In standard sampling workflows, GFMs aim to progressively transform the noisy graph $G_0$ into a clean graph $G_1$, which is a de novo generation process. However, real-world applications require graphs with specific properties, which may only exist within a small region of the entire generative space. In this case, de novo graph generation will result in invalid or low-quality graphs, which cannot effectively explore the high-potential region in the chemical space.

To address this issue, we propose a refinement strategy to explore the potential region by iteratively renoising and re-generating promising graphs, as shown in Figure 3. Specifically, we maintain a priority pool $\mathcal{B}$ to record graphs with top-$M$ reward scores. In each iteration, these graphs undergo a refinement cycle, consisting of two parts:

- **Renoising.** For a candidate graph $G_1 \in \mathcal{B}$, we revert it to an intermediate noisy state at time $t_\epsilon \in (0, 1)$ based on the conditional probability path:

$$p_{t_\epsilon|1}(z_{t_\epsilon}|z_1) = t_\epsilon \cdot \delta(z_{t_\epsilon}, z_1) + (1 - t_\epsilon) \cdot p_0(z_{t_\epsilon}). \quad (16)$$

Here, $t_\epsilon$ controls the perturbation level: a larger $t_\epsilon$ means a smaller noise amplitude, thus enabling controlled exploration. In practice, we typically sample multiple noisy graphs, $G_{t_\epsilon} \sim p(z_{t_\epsilon}|z_1)$, to explore the potential of $G_1$.

- **Regeneration.** Given a set of noisy graphs, we re-execute the denoising process of GFMs to generate more candidates. Subsequently, we evaluate the reward score of new graphs, compare them with the graphs in the priority queue, and retain $M$ graphs with the highest scores.

More technical details can be seen in Appendix B.

*Table 1.* Graph generation performance on two synthetic datasets: Planar and Tree. Results are reported as mean $\pm$ standard deviation over five sampling runs, each generating 40 graphs.

| Model | Step | Planar | | Tree | |
|---|---|---|---|---|---|
| | | V.U.N. $\uparrow$ | Ratio $\downarrow$ | V.U.N. $\uparrow$ | Ratio $\downarrow$ |
| Train set | - | 100 | 1.0 | 100 | 1.0 |
| GraphRNN | - | 0.0 | 490.2 | 0.0 | 607.0 |
| BiGG | - | 5.0 | 16.0 | 75.0 | 5.2 |
| GraphGen | - | 7.5 | 210.3 | 95.0 | 33.2 |
| CatFlow | - | 80.0 | - | - | - |
| DiGress | 1,000 | 77.5 | 5.1 | 90.0 | **1.6** |
| GBD | 1,000 | 92.5 | 7.7 | - | - |
| DisCo | 1,000 | $83.6_{\pm 2.1}$ | - | - | - |
| GDPO | 1,000 | $73.8_{\pm 2.5}$ | - | - | - |
| DeFoG | 50 | $\mathbf{95.0}_{\pm 3.2}$ | $3.2_{\pm 1.1}$ | $73.5_{\pm 9.0}$ | $2.5_{\pm 1.0}$ |
| Graph-GRPO | 50 | $\mathbf{95.0}_{\pm 4.0}$ | $\mathbf{1.5}_{\pm 0.5}$ | $\mathbf{97.5}_{\pm 1.6}$ | $2.2_{\pm 0.8}$ |

## 4. Experiment

### 4.1. Experimental Setup

Our framework builds upon DeFoG (Qin et al., 2025), which employs a graph Transformer equipped with Relative Random Walk Probabilities (RRWP) (Ma et al., 2023; Siraudin et al., 2025) as the denoising network. We benchmark the graph generation methods on three tasks: general graph generation on synthetic datasets, molecular optimization for protein docking, and target property optimization. The base model is pretrained on specific datasets for general graph generation and on ZINC250k (Irwin et al., 2012) for molecular optimization. If a checkpoint is provided, we download it from the repository[1]; otherwise, we train DeFoG by ourselves. See Appendix C for more implementation details.

### 4.2. General Graph Generation

**Setup.** We evaluate general graph generation on two synthetic benchmarks: *Planar* and *Tree* (Bergmeister et al., 2024). Both datasets consist of graphs with a fixed number of 64 nodes. Each dataset is split into 128 training graphs, 32 validation graphs, and 40 test graphs. At test time, we conduct five evaluations, generating 40 graphs each time, and report the mean and standard deviation. The generation quality is reflected in two metrics: Valid-Unique-Novel (V.U.N.) and the ratio over the training set.

**Reward Design.** We design the reward function as a combination of a hard validity constraint and a sum of the soft distribution matching metrics:

$$R(G) = \mathbb{I}(G) \cdot \left( \alpha + \frac{1-\alpha}{|\mathcal{K}|} \sum_{k \in \mathcal{K}} S_k \right), \quad (17)$$

where $\mathbb{I}(G) = 1$ if $G$ is valid, otherwise 0. The parameter

---

[1] https://github.com/manuelmlmadeira/DeFoG

*Table 2.* Experimental results on the protein docking task on ZINC250k (Irwin et al., 2012). Baseline results are taken from GDPO (Liu et al., 2024). We use the same benchmark evaluation protocol for a fair comparison. **The best results are highlighted in bold**.

| Target | Metric | RL-optimized Generative Models | | | | | | Basic Generative Models | | |
|---|---|---|---|---|---|---|---|---|---|---|
| | | Graph-GRPO | GDPO | DDPO | FREED | REINVENT | GCPN | DiGress-G | DiGress | MOOD |
| *parp1* | DS (top 5%) $\downarrow$ | **-12.515**$_{\pm 0.024}$ | -10.938$_{\pm 0.042}$ | -9.247$_{\pm 0.242}$ | -10.579$_{\pm 0.104}$ | -8.702$_{\pm 0.523}$ | -8.102$_{\pm 0.105}$ | -9.463$_{\pm 0.524}$ | -9.219$_{\pm 0.078}$ | -10.865$_{\pm 0.113}$ |
| | Hit Ratio $\uparrow$ | **60.763**$_{\pm 0.471}$ | 9.814$_{\pm 1.352}$ | 0.419$_{\pm 0.280}$ | 4.627$_{\pm 0.727}$ | 0.480$_{\pm 0.344}$ | 0 | 1.172$_{\pm 0.672}$ | 0.366$_{\pm 0.146}$ | 7.017$_{\pm 0.428}$ |
| *fa7* | DS (top 5%) $\downarrow$ | **-9.099**$_{\pm 0.053}$ | -8.691$_{\pm 0.074}$ | -7.739$_{\pm 0.244}$ | -8.378$_{\pm 0.044}$ | -7.205$_{\pm 0.264}$ | -6.688$_{\pm 0.186}$ | -7.318$_{\pm 0.213}$ | -7.736$_{\pm 0.156}$ | -8.160$_{\pm 0.071}$ |
| | Hit Ratio $\uparrow$ | **9.367**$_{\pm 0.753}$ | 3.449$_{\pm 0.188}$ | 0.342$_{\pm 0.685}$ | 1.332$_{\pm 0.113}$ | 0.213$_{\pm 0.081}$ | 0 | 0.321$_{\pm 0.370}$ | 0.182$_{\pm 0.232}$ | 0.733$_{\pm 0.141}$ |
| *5ht1b* | DS (top 5%) $\downarrow$ | **-11.399**$_{\pm 0.026}$ | -11.304$_{\pm 0.093}$ | -9.488$_{\pm 0.287}$ | -10.714$_{\pm 0.183}$ | -8.770$_{\pm 0.316}$ | -8.544$_{\pm 0.505}$ | -8.971$_{\pm 0.395}$ | -9.280$_{\pm 0.198}$ | -11.145$_{\pm 0.042}$ |
| | Hit Ratio $\uparrow$ | **46.634**$_{\pm 1.042}$ | 34.359$_{\pm 2.734}$ | 5.488$_{\pm 1.989}$ | 16.767$_{\pm 0.897}$ | 2.453$_{\pm 0.561}$ | 1.455$_{\pm 1.173}$ | 2.821$_{\pm 1.140}$ | 4.236$_{\pm 0.887}$ | 18.673$_{\pm 0.423}$ |
| *braf* | DS (top 5%) $\downarrow$ | -11.141$_{\pm 0.010}$ | **-11.197**$_{\pm 0.132}$ | -9.470$_{\pm 0.373}$ | -10.561$_{\pm 0.080}$ | -8.392$_{\pm 0.400}$ | -8.713$_{\pm 0.155}$ | -8.825$_{\pm 0.459}$ | -9.052$_{\pm 0.044}$ | -11.063$_{\pm 0.034}$ |
| | Hit Ratio $\uparrow$ | **10.041**$_{\pm 0.496}$ | 9.039$_{\pm 1.473}$ | 0.445$_{\pm 0.297}$ | 2.940$_{\pm 0.359}$ | 0.127$_{\pm 0.088}$ | 0 | 0.152$_{\pm 0.303}$ | 0.122$_{\pm 0.141}$ | 5.240$_{\pm 0.285}$ |
| *jak2* | DS (top 5%) $\downarrow$ | **-11.123**$_{\pm 0.025}$ | -10.183$_{\pm 0.124}$ | -8.990$_{\pm 0.221}$ | -9.735$_{\pm 0.022}$ | -8.165$_{\pm 0.277}$ | -8.073$_{\pm 0.093}$ | -8.360$_{\pm 0.217}$ | -8.706$_{\pm 0.222}$ | -10.147$_{\pm 0.060}$ |
| | Hit Ratio $\uparrow$ | **52.897**$_{\pm 0.245}$ | 13.405$_{\pm 1.515}$ | 1.717$_{\pm 0.684}$ | 5.800$_{\pm 0.295}$ | 0.613$_{\pm 0.167}$ | 0 | 0.311$_{\pm 0.621}$ | 0.861$_{\pm 0.332}$ | 9.200$_{\pm 0.524}$ |

$\alpha \in (0, 1)$ represents a base reward allocated for valid graph, while the remaining portion $(1 - \alpha)$ serves as a quality bonus for structural similarities, *i.e.*, $\mathcal{K} = \{\text{deg}, \text{clus}, \text{orb}\}$, between the generated graphs and training graphs. In our experiments, we set $\alpha = 0.65$ to prioritize validity while reserving significant capacity for structural refinement.

**Result.** Table 1 presents the graph generation results on two synthetic datasets. In the Planar dataset, Graph-GRPO has a lower ratio than the base model, demonstrating better structural alignment without degrading the V.U.N. score. In the Tree dataset, the improvement is even more pronounced: Graph-GRPO boosts V.U.N. from 73.5% to 97.5% while simultaneously reducing the ratio. Moreover, with only 50 denoising steps, Graph-GRPO outperforms graph diffusion models with 1,000 steps, *e.g.*, DiGress (Vignac et al., 2023), GBD (Liu et al., 2025), and DisCo (Xu et al., 2024), as well as the graph diffusion policy optimization (GDPO) (Liu et al., 2024), validating the effectiveness of our framework. More baselines and results are provided in Appendix G.2.

### 4.3. Protein Docking

**Setup.** We further evaluate graph generation methods on the protein docking application, which aims to generate molecules that dock to specific proteins, and possess desirable properties, such as novelty (*NOV*), drug-likeness (*QED*), and synthetic accessibility (*SA*). We follow the setup in GDPO (Liu et al., 2024), which uses the hit ratio (%) and the top-5% docking score (DS) as metrics, and considers five target proteins: *parp1*, *fa7*, *5ht1b*, *braf* and *jak2*. Hit ratio is defined as the proportion of effective molecules with *NOV* > 0.6, *QED* > 0.5, and *SA* < 5. Docking score indicates the negative binding affinity with specific proteins. We compare Graph-GRPO against several representative baselines, including RL-optimized models, *e.g.*, GDPO (Liu et al., 2024), DDPO (Black et al., 2024), FREED (Yang et al., 2021), REINVENT (Olivecrona et al., 2017), and GCPN (You et al., 2018a), and basic graph generative models, *e.g.*, DiGress (Vignac et al., 2023) with and without

guidance, and MOOD (Lee et al., 2023).

**Reward Design.** We adopt the same reward as GDPO, which uses a composite reward function to balance binding affinity, chemical quality, and novelty, defined as

$$R(G) = 0.1 \cdot (R_{\text{QED}} + R_{\text{SA}}) + 0.3 \cdot R_{\text{Nov}} + 0.5 \cdot R_{\text{DS}}. \quad (18)$$

Further details are provided in Appendix F.1.

**Result.** Table 2 presents results on the protein docking task. We have two observations: First, Graph-GRPO obtains the optimal or sub-optimal docking scores on the five target proteins, demonstrating its effectiveness in synthesizing molecules with high binding affinity. Second, Graph-GRPO exhibits better sampling efficiency, reflected in its higher hit ratio compared to other methods. For example, given the *parp1* protein, molecules generated by Graph-GRPO have a 60% hit ratio, which is 6× higher than the best baseline GDPO. This result indicates that Graph-GRPO can effectively align GFMs with task-specific objectives and efficiently explore the high-potential region in the chemical space. As we limit the models to generate up to 2,048 molecules, a higher hit ratio also benefits the docking score because it uses more candidates for evaluation. Detailed experimental results are provided in Appendix G.1.

### 4.4. Target Property Optimization

**Setup.** We finally use the Practical Molecular Optimization (PMO) benchmark (Gao et al., 2022) to evaluate the ability to generate molecules with target properties. The PMO benchmark has 23 diverse tasks and evaluates the generated molecules under a strict budget of 10,000 oracle calls. We consider two experimental setups: (1) **Prescreening**, where a high-quality initial pool is first constructed by prescreening the ZINC250k dataset. This process will consume 250k oracle calls, but it provides a good starting point. (2) **Cold-Start (w/o Prescreening)**, which uses de novo generation without any prior candidate pool. We compare Graph-GRPO against several baselines, including fragment-based, *e.g.*, InVirtuoGen (Kaech et al., 2025), GenMol (Lee

*Table 3.* Target property optimization results on the PMO benchmark (Gao et al., 2022) (AUC-top10). Baseline results are taken from InVirtuoGen (Kaech et al., 2025) and GenMol (Lee et al., 2025). Results are shown for both prescreening and cold-start settings, **with best scores highlighted in bold**.

| Oracle | w/ Prescreening | | | | Cold-Start (w/o Prescreening) | | | | |
|---|---|---|---|---|---|---|---|---|---|
| | **Graph-GRPO** | **InVirtuoGen** | **GenMol** | **f-RAG** | **Graph-GRPO** | **InVirtuoGen** | **Gen. GFN** | **Mol GA** | **REINVENT** |
| Albuterol Similarity | **0.994**$_{\pm0.000}$ | 0.975$_{\pm0.016}$ | 0.937$_{\pm0.010}$ | 0.977$_{\pm0.002}$ | **0.994**$_{\pm0.000}$ | 0.950$_{\pm0.017}$ | 0.949$_{\pm0.010}$ | 0.896$_{\pm0.035}$ | 0.882$_{\pm0.006}$ |
| Amlodipine MPO | 0.823$_{\pm0.008}$ | **0.836**$_{\pm0.031}$ | 0.810$_{\pm0.012}$ | 0.749$_{\pm0.019}$ | **0.823**$_{\pm0.008}$ | 0.733$_{\pm0.043}$ | 0.761$_{\pm0.019}$ | 0.688$_{\pm0.039}$ | 0.635$_{\pm0.035}$ |
| Celecoxib Rediscovery | **0.890**$_{\pm0.000}$ | 0.839$_{\pm0.013}$ | 0.826$_{\pm0.018}$ | 0.778$_{\pm0.007}$ | **0.890**$_{\pm0.000}$ | 0.798$_{\pm0.028}$ | 0.802$_{\pm0.029}$ | 0.567$_{\pm0.083}$ | 0.713$_{\pm0.067}$ |
| Deco Hop | 0.942$_{\pm0.005}$ | **0.968**$_{\pm0.012}$ | 0.960$_{\pm0.010}$ | 0.936$_{\pm0.011}$ | **0.762**$_{\pm0.127}$ | 0.748$_{\pm0.109}$ | 0.733$_{\pm0.109}$ | 0.649$_{\pm0.025}$ | 0.666$_{\pm0.044}$ |
| DRD2 | **0.995**$_{\pm0.000}$ | **0.995**$_{\pm0.000}$ | **0.995**$_{\pm0.000}$ | 0.992$_{\pm0.000}$ | **0.995**$_{\pm0.000}$ | 0.985$_{\pm0.002}$ | 0.974$_{\pm0.006}$ | 0.936$_{\pm0.016}$ | 0.945$_{\pm0.007}$ |
| Fexofenadine MPO | **0.984**$_{\pm0.001}$ | 0.904$_{\pm0.000}$ | 0.894$_{\pm0.028}$ | 0.856$_{\pm0.016}$ | **0.984**$_{\pm0.001}$ | 0.845$_{\pm0.016}$ | 0.856$_{\pm0.039}$ | 0.825$_{\pm0.019}$ | 0.784$_{\pm0.006}$ |
| GSK3b | 0.948$_{\pm0.015}$ | **0.988**$_{\pm0.001}$ | 0.986$_{\pm0.003}$ | 0.969$_{\pm0.003}$ | **0.965**$_{\pm0.006}$ | 0.952$_{\pm0.016}$ | 0.881$_{\pm0.042}$ | 0.843$_{\pm0.039}$ | 0.865$_{\pm0.043}$ |
| Isomers C7H8N2O2 | **0.995**$_{\pm0.000}$ | 0.988$_{\pm0.002}$ | 0.942$_{\pm0.004}$ | 0.955$_{\pm0.008}$ | **0.995**$_{\pm0.000}$ | 0.968$_{\pm0.005}$ | 0.969$_{\pm0.003}$ | 0.878$_{\pm0.026}$ | 0.852$_{\pm0.036}$ |
| Isomers C9H10N2O2PF2Cl | **0.932**$_{\pm0.011}$ | 0.898$_{\pm0.018}$ | 0.833$_{\pm0.014}$ | 0.850$_{\pm0.005}$ | **0.932**$_{\pm0.011}$ | 0.874$_{\pm0.013}$ | 0.897$_{\pm0.007}$ | 0.865$_{\pm0.012}$ | 0.642$_{\pm0.054}$ |
| JNK3 | **0.910**$_{\pm0.019}$ | 0.898$_{\pm0.031}$ | 0.906$_{\pm0.023}$ | 0.904$_{\pm0.004}$ | **0.910**$_{\pm0.019}$ | 0.825$_{\pm0.016}$ | 0.764$_{\pm0.069}$ | 0.702$_{\pm0.123}$ | 0.783$_{\pm0.023}$ |
| Median 1 | **0.388**$_{\pm0.000}$ | 0.386$_{\pm0.003}$ | **0.398**$_{\pm0.000}$ | 0.340$_{\pm0.007}$ | **0.388**$_{\pm0.000}$ | 0.342$_{\pm0.008}$ | 0.379$_{\pm0.010}$ | 0.257$_{\pm0.009}$ | 0.356$_{\pm0.009}$ |
| Median 2 | 0.322$_{\pm0.039}$ | **0.377**$_{\pm0.006}$ | 0.359$_{\pm0.004}$ | 0.323$_{\pm0.005}$ | 0.300$_{\pm0.002}$ | 0.288$_{\pm0.008}$ | 0.294$_{\pm0.007}$ | **0.301**$_{\pm0.021}$ | 0.276$_{\pm0.008}$ |
| Mestranol Similarity | 0.980$_{\pm0.002}$ | **0.991**$_{\pm0.002}$ | 0.982$_{\pm0.006}$ | 0.671$_{\pm0.021}$ | **0.974**$_{\pm0.002}$ | 0.797$_{\pm0.033}$ | 0.708$_{\pm0.057}$ | 0.591$_{\pm0.053}$ | 0.618$_{\pm0.048}$ |
| Osimertinib MPO | **0.924**$_{\pm0.003}$ | 0.881$_{\pm0.012}$ | 0.876$_{\pm0.008}$ | 0.866$_{\pm0.009}$ | **0.922**$_{\pm0.002}$ | 0.870$_{\pm0.005}$ | 0.860$_{\pm0.008}$ | 0.844$_{\pm0.015}$ | 0.837$_{\pm0.009}$ |
| Perindopril MPO | 0.690$_{\pm0.033}$ | **0.753**$_{\pm0.019}$ | 0.718$_{\pm0.012}$ | 0.681$_{\pm0.017}$ | **0.689**$_{\pm0.036}$ | 0.645$_{\pm0.032}$ | 0.595$_{\pm0.014}$ | 0.547$_{\pm0.022}$ | 0.537$_{\pm0.016}$ |
| QED | **0.944**$_{\pm0.000}$ | 0.943$_{\pm0.000}$ | 0.942$_{\pm0.000}$ | 0.939$_{\pm0.001}$ | **0.944**$_{\pm0.000}$ | 0.942$_{\pm0.000}$ | 0.942$_{\pm0.000}$ | 0.941$_{\pm0.001}$ | 0.941$_{\pm0.000}$ |
| Ranolazine MPO | **0.928**$_{\pm0.001}$ | 0.854$_{\pm0.012}$ | 0.821$_{\pm0.011}$ | 0.820$_{\pm0.016}$ | **0.928**$_{\pm0.001}$ | 0.848$_{\pm0.010}$ | 0.819$_{\pm0.018}$ | 0.804$_{\pm0.011}$ | 0.760$_{\pm0.009}$ |
| Scaffold Hop | 0.711$_{\pm0.113}$ | **0.711**$_{\pm0.081}$ | 0.628$_{\pm0.008}$ | 0.576$_{\pm0.014}$ | **0.622**$_{\pm0.015}$ | 0.589$_{\pm0.021}$ | 0.615$_{\pm0.100}$ | 0.527$_{\pm0.025}$ | 0.560$_{\pm0.019}$ |
| Sitagliptin MPO | **0.879**$_{\pm0.014}$ | 0.743$_{\pm0.022}$ | 0.584$_{\pm0.034}$ | 0.601$_{\pm0.011}$ | **0.879**$_{\pm0.014}$ | 0.709$_{\pm0.029}$ | 0.634$_{\pm0.039}$ | 0.582$_{\pm0.040}$ | 0.021$_{\pm0.003}$ |
| Thiothixene Rediscovery | **0.842**$_{\pm0.001}$ | 0.652$_{\pm0.024}$ | 0.692$_{\pm0.123}$ | 0.584$_{\pm0.009}$ | **0.842**$_{\pm0.001}$ | 0.625$_{\pm0.014}$ | 0.583$_{\pm0.034}$ | 0.519$_{\pm0.041}$ | 0.534$_{\pm0.013}$ |
| Troglitazone Rediscovery | 0.711$_{\pm0.008}$ | 0.853$_{\pm0.003}$ | **0.867**$_{\pm0.022}$ | 0.448$_{\pm0.017}$ | **0.711**$_{\pm0.008}$ | 0.595$_{\pm0.053}$ | 0.511$_{\pm0.054}$ | 0.427$_{\pm0.031}$ | 0.441$_{\pm0.032}$ |
| Valsartan SMARTS | 0.841$_{\pm0.019}$ | **0.935**$_{\pm0.012}$ | 0.822$_{\pm0.042}$ | 0.627$_{\pm0.058}$ | **0.841**$_{\pm0.019}$ | 0.210$_{\pm0.297}$ | 0.135$_{\pm0.271}$ | 0.000$_{\pm0.000}$ | 0.178$_{\pm0.358}$ |
| Zaleplon MPO | **0.697**$_{\pm0.004}$ | 0.624$_{\pm0.040}$ | 0.584$_{\pm0.011}$ | 0.486$_{\pm0.004}$ | **0.697**$_{\pm0.004}$ | 0.536$_{\pm0.006}$ | 0.552$_{\pm0.033}$ | 0.519$_{\pm0.029}$ | 0.358$_{\pm0.062}$ |
| **Sum (AUC-top10)** | **19.270** | 18.993 | 18.362 | 16.928 | **18.987** | 16.676 | 16.213 | 14.708 | 14.184 |

*Table 4.* Ablation studies on the PMO benchmark. We report the averaged AUC-top10 over all 23 tasks. All oracle calls used by refinement are counted under the same 10k budget.

| Method | RL | Refine-ment | Pre-screening | Dynamic Prior | AUC-top10 |
|---|---|---|---|---|---|
| DeFoG | - | - | - | - | 11.079 |
| DeFoG | - | ✓ | - | - | 15.251 |
| DeFoG | - | ✓ | ✓ | - | 15.887 |
| Graph-GRPO | ✓ | - | - | - | 16.901 |
| Graph-GRPO | ✓ | - | - | ✓ | 17.450 |
| Graph-GRPO | ✓ | ✓ | - | - | 17.950 |
| Graph-GRPO | ✓ | ✓ | - | ✓ | 18.987 |
| Graph-GRPO | ✓ | ✓ | ✓ | - | 18.033 |
| Graph-GRPO | ✓ | ✓ | ✓ | ✓ | **19.270** |

et al., 2025), REINVENT (Olivecrona et al., 2017), Genetic GFN (Kim et al., 2024), and f-RAG (Lee et al., 2024), and graph-based, *e.g.*, Mol GA (Tripp & Hernández-Lobato, 2023). It is worth noting that Graph-GRPO builds upon DeFoG as the base model, which is pretrained solely on the ZINC250k dataset. In contrast, some baseline methods, such as InVirtuoGen and GenMol, are pretrained on significantly larger datasets (Irwin et al., 2012; Chambers et al., 2013) containing over one billion molecules.

**Reward Design.** We directly adopt the standard oracle scores defined by the benchmark protocol as our reward. Implementation details are provided in Appendix F.2.

**Results.** Table 3 presents the results on PMO benchmark. In the cold-start setting, Graph-GRPO outperforms baselines by a clear margin and matches the performance of meth-ods that rely on expensive prescreening. The improvement over baselines mainly comes from some hard tasks, such as Thiothixene Rediscovery and Troglitazone Rediscovery, indicating that Graph-GRPO can effectively explore the high-potential regions in chemical space, even without any prior knowledge. In the prescreening setting, Graph-GRPO further advances its performance to a new state-of-the-art with an AUC of 19.270, validating the effectiveness of both prescreening and refinement. Notably, all oracle calls during refinement are fully counted toward the total budget. See a detailed explanation in Appendix C.3.

### 4.5. Ablation Study

We conduct a full component ablation on the PMO benchmark to study the effects of components in Graph-GRPO: RL training, refinement, prescreening, and dynamic prior. As shown in Table 4, all variants are evaluated by the averaged AUC-top10 over 23 PMO tasks under the same 10k oracle-call budget.

RL training is the dominant contributor, improving the AUC-top10 from 11.079 to 16.901 by optimizing the task-specific reward. Refinement brings substantial gains both with and without RL, showing the benefit of iterative renoising and regeneration in exploring high-potential regions. Prescreening provides a smaller but consistent improvement, since RL and refinement already guide the model toward high-reward candidates. Dynamic prior further improves performance by adapting the sampling prior to high-scoring regions, and the full Graph-GRPO achieves the best AUC-top10 of 19.270.

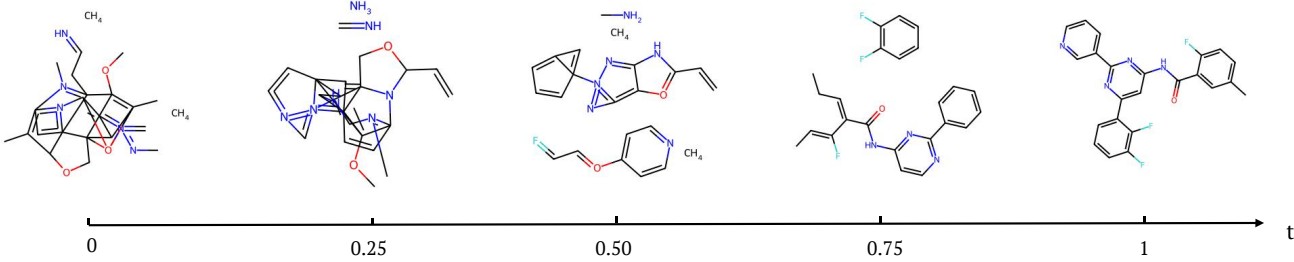

*Figure 4.* Visualization of the sampling trajectory. From left to right, the graphs $G_t$ move from a prior distribution ($t = 0$) to the data distribution ($t = 1$). Visualizations of the predicted clean state $\hat{z}_1$ at the corresponding time steps are provided in Figure 8.

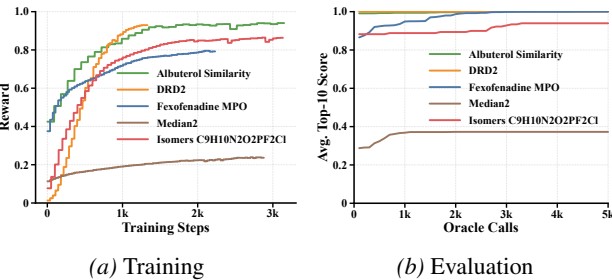

|(a) Training|(b) Evaluation|
|---|---|

*Figure 5.* Visualization of training and evaluation curves in Graph-GRPO. We select five representative tasks from the PMO benchmark. (a) Training reward curves plotted against training steps. (b) Average top-10 scores plotted against the number of oracle calls. For brevity, we only visualize the first 5,000 oracle calls.

To further examine the influence of factors beyond the main component ablation, we provide detailed studies in the appendix. Specifically, we analyze the effect of the renoising time $t_\epsilon$ in Appendix E.1, compare Graph-GRPO with a REINFORCE variant without analytic transition in Appendix E.2, and evaluate the scalability of Graph-GRPO under different graph sizes and denoising steps in Appendix E.3.

### 4.6. Visualization

Figure 4 visualizes the denoising trajectory on the Deco Hop task, illustrating how Graph-GRPO progressively transforms an initially disordered and noisy graph into a chemically valid, high-quality molecule that satisfies pharmacophore constraints. Figure 5a highlights the stability of the training process, where the reward score gradually increases with the number of training steps. Moreover, Figure 5b shows the average top-10 scores with the number of oracle calls in the evaluation stage. It can be observed that the scores are higher than the training stage, indicating the effectiveness of the refinement strategy.

## 5. Related Work

**Graph Generative Models.** Graph generation can be roughly divided into autoregressive, one-shot, and diffusion

or flow-based models. Autoregressive models construct graphs sequentially (e.g., GraphRNN (You et al., 2018b), BiGG (Dai et al., 2020), GraphGen (Goyal et al., 2020)), while one-shot methods generate graphs holistically via latent-variable or adversarial models (e.g., JT-VAE (Jin et al., 2018), MolGAN (Cao & Kipf, 2018), SPECTRE (Martinkus et al., 2022)). Diffusion-based methods have evolved from continuous scoremodels (e.g., GDSS (Jo et al., 2022)) to discrete diffusion (e.g., DiGress (Vignac et al., 2023), DisCo (Xu et al., 2024)). Recently, discrete flow matching formulates graph generation as CTMC dynamics, e.g., Cat-Flow (Eijkelboom et al., 2024), DeFoG (Qin et al., 2025).

**Goal-Directed Molecular Generation and Optimization.** Early works for molecular generation under task-specific objectives apply RL to sequential molecule construction, e.g., GCPN (You et al., 2018a), REINVENT (Olivecrona et al., 2017). Evolutionary search provides an alternative paradigm (e.g., Mol GA (Tripp & Hernández-Lobato, 2023)). Recent methods integrate optimization objectives with deep generative models. In structure-based settings, diffusion-style generators are optimized with docking rewards, including GDPO (Liu et al., 2024), DDPO (Black et al., 2024), and FREED (Yang et al., 2021). Some other systems emphasize candidate reuse and local modification, including InVirtuoGen (Kaech et al., 2025), GenMol (Lee et al., 2025), f-RAG (Lee et al., 2024), Genetic GFN (Kim et al., 2024).

**Molecule and Protein Representation Learning.** Since our work is related to molecule and protein modeling, for completeness we briefly review some works. They primarily learn molecule or protein embeddings in a self-supervised method, and are fine-tuned for downstream tasks (Wang et al., 2022; Jaeger et al., 2018; Li & Jiang, 2021; Brandes et al., 2022; Zhou et al., 2023; Zhang et al., 2022). However, they are not graph generation methods. On the contrary, our work focuses on graph generation using online RL.

## 6. Conclusion

We propose Graph-GRPO, an online RL framework for optimizing GFMs with task-specific rewards. By deriving analytic transition probabilities, Graph-GRPO enables

fully differentiable rollouts and integrates GFMs with RL training. A refinement strategy is proposed to refine high-reward samples through localized exploration and prevent the generation of invalid or low-quality graphs. Experiments demonstrate that Graph-GRPO consistently improves generation quality and achieves state-of-the-art performance. Our results highlight a principled pathway for aligning GFMs with complex downstream objectives. A promising future work is to apply Graph-GRPO to broader downstream applications, such as material generation.

## Acknowledgment

This work is supported in part by the National Natural Science Foundation of China (No. 62322203).

## Impact Statement

This paper presents work whose goal is to advance the field of Graph Machine Learning. There are many potential societal consequences of our work, none of which we feel must be specifically highlighted here.

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

# A. Derivation of Discrete Flow Matching for Graphs

### A.1. Factorized Probability Path

As defined in Section 2, the graph state $z_t$ consists of node features $X$ and edge features $E$. We assume the generative distribution factorizes over the dimensions of the graph. Let $z_t$ denote the discrete variable at a specific dimension (representing either a node or an edge feature), and $z_{t+dt}$ denote a target state at that dimension.

For a single dimension, we define the conditional probability path $p_{t|1}(z_t \mid z_1)$ as a linear interpolation between a non-uniform prior $p_0$ and the clean data $z_1$:

$$p_{t|1}(z_t \mid z_1) = t \cdot \delta(z_t, z_1) + (1 - t) \cdot p_0(z_t), \tag{19}$$

where $p_0(z_t)$ is the prior probability of state $z_t$. We assume $p_0$ has full support.

### A.2. Derivation of the Ideal Conditional Rate $R_t$

To simulate this probability path using a Continuous-Time Markov Chain (CTMC), we require a transition rate matrix $R_t^*$ that satisfies the continuity equation. We focus on the transition rate function (a specific entry of the matrix) from the current state $z_t$ to a target candidate state $z_{t+dt}$ (where $z_{t+dt} \neq z_t$).

**1. Time Derivative and Flux.** The time derivative of the probability path is:

$$\partial_t p_{t|1}(z_{t+dt} \mid z_1) = \delta(z_{t+dt}, z_1) - p_0(z_{t+dt}). \tag{20}$$

The net probability flux difference $\Delta$ between the target state $z_{t+dt}$ and the current state $z_t$ is defined as:

$$\Delta(z_t \to z_{t+dt}) := \partial_t p_{t|1}(z_{t+dt} \mid z_1) - \partial_t p_{t|1}(z_t \mid z_1)$$
$$= \Big(\delta(z_{t+dt}, z_1) - \delta(z_t, z_1)\Big) + \Big(p_0(z_t) - p_0(z_{t+dt})\Big). \tag{21}$$

**2. Rate Construction.** Following the formulation in discrete flow matching literature (e.g., DeFoG), the transition rate is defined as the rectified flux normalized by the current probability mass and the count of reachable states:

$$R_t(z_t, z_{t+dt} \mid z_1) = \frac{\text{ReLU}\big(\Delta(z_t \to z_{t+dt})\big)}{Z_t^{>0} \cdot p_{t|1}(z_t \mid z_1)}, \tag{22}$$

where $Z_t^{>0} = |\{z_t : p_{t|1}(z_t \mid z_1) > 0\}|$ denotes the number of states with non-zero probability at time $t$. Note that under the assumption that the prior $p_0$ has full support, $p_{t|1}(z_t \mid z_1)$ remains positive for all states throughout the process ($t < 1$). Thus, $Z_t^{>0}$ is constant and equal to the total number of categories $S$ (i.e., $Z_t^{>0} \equiv S$).

We analyze the rate in the active flow regime where the current state has not yet reached the clean state (i.e., $z_t \neq z_1$). In this case, $\delta(z_t, z_1) = 0$, and the denominator simplifies to $Z_t^{>0}(1 - t)p_0(z_t)$. The rate splits into two cases:

- **Case A: Target is the Clean Data ($z_{t+dt} = z_1$).**

$$R_t(z_t, z_{t+dt} \mid z_1) = \frac{1 + p_0(z_t) - p_0(z_1)}{Z_t^{>0}(1 - t)p_0(z_t)}. \tag{23}$$

  *Note: The ReLU is omitted here because $1 + p_0(z_t) - p_0(z_1) \geq 0$ holds for any valid probability distribution.*

- **Case B: Target is NOT the Clean Data ($z_{t+dt} \neq z_1$).**

$$R_t(z_t, z_{t+dt} \mid z_1) = \frac{\text{ReLU}(p_0(z_t) - p_0(z_{t+dt}))}{Z_t^{>0}(1 - t)p_0(z_t)}. \tag{24}$$

## A.3. Derivation of the Learned Rate $R_t^\theta$

Since the clean graph $z_1$ is unknown during generation, we parameterize the transitions using the expected rate over the posterior distribution predicted by the network $f_\theta$.

Let $p_\theta(z_1 = \hat{z}_1 \mid z_t)$ denote the probability assigned by the network that the true clean state is category $\hat{z}_1$, given the current noisy state $z_t$. The parameterized rate $R_t^\theta(z_t, z_{t+\mathrm{d}t})$ is defined as the expectation:

$$R_t^\theta(z_t, z_{t+\mathrm{d}t}) = \mathbb{E}_{\hat{z}_1 \sim p_\theta(\cdot \mid z_t)}\left[R_t(z_t, z_{t+\mathrm{d}t} \mid \hat{z}_1)\right], \tag{25}$$

Expanding the expectation by summing over all possible clean states $\hat{z}_1$, we group the terms into three categories based on the relationship between the clean state $\hat{z}_1$, the current state $z_t$, and the destination $z_{t+\mathrm{d}t}$:

$$
\begin{aligned}
R_t^\theta(z_t, z_{t+\mathrm{d}t}) = {} & \underbrace{p_\theta(z_t \mid z_t) \cdot 0}_{\text{Term 1: } \hat{z}_1 = z_t \text{ (Stability)}} \\
& + \underbrace{p_\theta(z_{t+\mathrm{d}t} \mid z_t) \cdot R_t(z_t, z_{t+\mathrm{d}t} \mid z_1 = z_{t+\mathrm{d}t})}_{\text{Term 2: } \hat{z}_1 = z_{t+\mathrm{d}t} \text{ (Target-Driven)}} \\
& + \sum_{\hat{z}_1 \notin \{z_t, z_{t+\mathrm{d}t}\}} \underbrace{p_\theta(\hat{z}_1 \mid z_t) \cdot R_t(z_t, z_{t+\mathrm{d}t} \mid z_1 = \hat{z}_1)}_{\text{Term 3: } \hat{z}_1 \text{ is other (Prior-Correction)}}.
\end{aligned} \tag{26}
$$

Substituting the closed-form solutions from the previous section (using Case A for Term 2 and Case B for Term 3) and applying the identity $\sum_{\hat{z}_1 \notin \{z_t, z_{t+\mathrm{d}t}\}} p_\theta(\hat{z}_1 \mid z_t) = 1 - p_\theta(z_t \mid z_t) - p_\theta(z_{t+\mathrm{d}t} \mid z_t)$, we obtain the fully parameterized expression:

$$R_t^\theta(z_t, z_{t+\mathrm{d}t}) = \frac{p_\theta(z_{t+\mathrm{d}t} \mid z_t)\big(1 + p_0(z_t) - p_0(z_{t+\mathrm{d}t})\big) + \big(1 - p_\theta(z_t \mid z_t) - p_\theta(z_{t+\mathrm{d}t} \mid z_t)\big)\mathrm{ReLU}\big(p_0(z_t) - p_0(z_{t+\mathrm{d}t})\big)}{Z_t^{>0}(1-t)p_0(z_t)}. \tag{27}$$

For notational brevity, we define the shorthand $p_\theta(z) \triangleq p_\theta(z \mid z_t)$ to denote the predicted probability of state $z$. Under this simplified notation, the final analytical transition rate is expressed as:

$$\boxed{R_t^\theta(z_t, z_{t+\mathrm{d}t}) = \frac{p_\theta(z_{t+\mathrm{d}t})\big(1 + p_0(z_t) - p_0(z_{t+\mathrm{d}t})\big) + \big(1 - p_\theta(z_t) - p_\theta(z_{t+\mathrm{d}t})\big)\mathrm{ReLU}\big(p_0(z_t) - p_0(z_{t+\mathrm{d}t})\big)}{Z_t^{>0}(1-t)p_0(z_t)}.} \tag{28}$$

**Diagonal Elements and Implementation.** The derivation above focuses on the transition rates for state changes ($z_{t+\mathrm{d}t} \neq z_t$). For theoretical completeness, the diagonal elements of the rate matrix are determined by the conservation of probability mass, satisfying $R_t^\theta(z_t, z_t) = -\sum_{z_{t+\mathrm{d}t} \neq z_t} R_t^\theta(z_t, z_{t+\mathrm{d}t})$.

In practical engineering implementation, specifically for the discrete-time sampling process with a time step $\Delta t$, we do not explicitly model the diagonal rates. Instead, we compute the off-diagonal transition probabilities using the analytical solution in Eq. 28 (approximated as $p(z_{t+\Delta t} \mid z_t) \approx R_t^\theta(z_t, z_{t+\Delta t})\Delta t$). The probability of remaining in the current state is then derived by subtracting the sum of off-diagonal probabilities from 1:

$$p(z_{t+\Delta t} = z_t \mid z_t) = 1 - \sum_{z_{t+\Delta t} \neq z_t} p(z_{t+\Delta t} \mid z_t). \tag{29}$$

This approach ensures that the transition probability distribution is always normalized (i.e., the row sum equals 1) and avoids numerical instability.

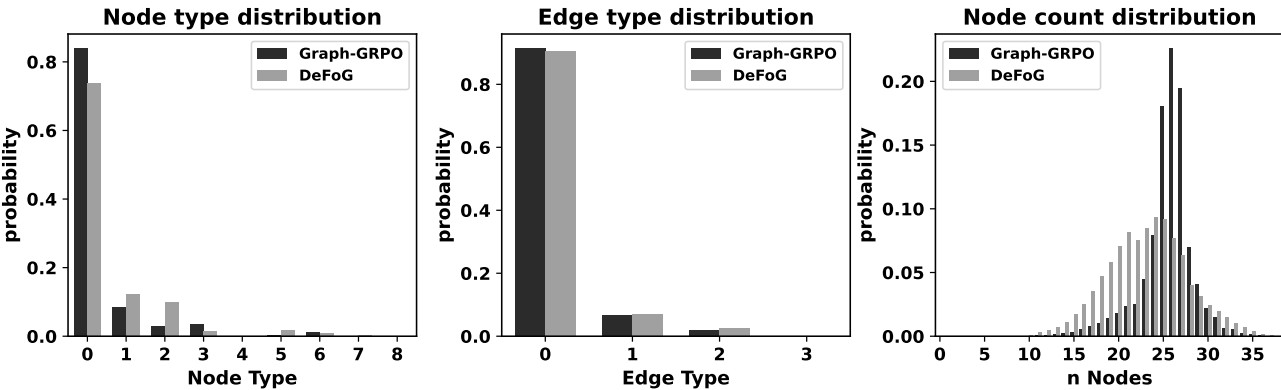

*Figure 6.* Comparison of learned prior distributions between the pre-trained DeFoG backbone and Graph-GRPO on the DRD2 task. The histograms illustrate the (Left) Node Type, (Middle) Edge Type, and (Right) Node Count distributions. Graph-GRPO dynamically aligns these priors with the characteristics of high-reward candidates, resulting in significant distributional shifts and sharper concentrations compared to the static baseline.

## B. Dynamic Prior and Graph Size Adjustment

In standard discrete flow matching, the prior distribution $p_0$ and the graph size distribution $p(N)$ are estimated from the training dataset and kept fixed during generation. However, under reinforcement learning, high-reward graphs discovered during training often exhibit task-specific structural patterns that differ substantially from the original data distribution. Using a fixed prior during sampling and refinement therefore leads to inefficient exploration and wasted oracle calls.

**Non-Parametric Prior Learning during Training.** During the *training phase*, Graph-GRPO maintains a non-parametric memory of high-reward samples and continuously estimates empirical distributions from these samples. This process does not rely on gradient-based optimization and is purely statistics-driven.

Specifically, we maintain a global buffer of high-performing graphs collected throughout RL training. From this buffer, we estimate empirical distributions over node types, edge types, and graph sizes. These empirical statistics are updated online and are *not* used to modify the denoising network parameters during training.

**Usage during Sampling** During the *sampling stage*, the learned non-parametric distributions are used to guide generation. Initial noise states and graph sizes are sampled from these distributions instead of the original dataset-derived priors. This adjustment is applied consistently to both de novo generation and all refinement steps, allowing the model to concentrate sampling on structural regions that have been empirically shown to yield high rewards.

### B.1. Global Reward Buffer

We maintain a global priority queue, denoted as the Global Reward Buffer $\mathcal{M}$, with a fixed capacity $K_{\mathcal{M}}$ (e.g., $K_{\mathcal{M}} = 1000$). Throughout training, any sampled trajectory $\tau$ with reward $R(\tau)$ exceeding a minimal threshold is treated as a candidate. At the end of each sampling phase, new candidates are merged into $\mathcal{M}$, and only the top-$K_{\mathcal{M}}$ unique graphs ranked by reward are retained. This buffer serves as the non-parametric source for estimating adaptive generative priors.

### B.2. Adaptive Prior Distribution ($p_0$)

The prior distribution $p_0 = \{p_0^X, p_0^E\}$ determines the initial noise state $z_0$. At the end of each training epoch, we compute empirical marginal distributions of node and edge types from the Global Reward Buffer $\mathcal{M}$, denoted as $\hat{p}_{\mathcal{M}}^X$ and $\hat{p}_{\mathcal{M}}^E$.

These empirical distributions are used to update the prior via an exponential moving average:

$$p_0^X \leftarrow (1 - \alpha)\, p_0^X + \alpha\, \hat{p}_{\mathcal{M}}^X, \tag{30}$$

$$p_0^E \leftarrow (1 - \alpha)\, p_0^E + \alpha\, \hat{p}_{\mathcal{M}}^E, \tag{31}$$

where $\alpha = 0.05$. This update is performed across training epochs and does not affect the continuous-time denoising dynamics within a single sampling trajectory.

### B.3. Adaptive Graph Size Distribution

We apply the same non-parametric strategy to the graph size distribution. From the graphs stored in $\mathcal{M}$, we construct an empirical size histogram $\hat{P}_{\mathcal{M}}(N)$ and update the sampling distribution as:

$$P(N) \leftarrow (1 - \alpha)\, P(N) + \alpha\, \hat{P}_{\mathcal{M}}(N). \tag{32}$$

During refinement, the adaptive size distribution is only used to initialize candidate graphs. Once a graph enters the localized refinement loop, its number of nodes is fixed to preserve the identified core scaffold. As illustrated in Figure 6, jointly adapting node types, edge types, and graph sizes enables Graph-GRPO to focus sampling on high-reward chemical subspaces and significantly improves sampling efficiency.

## C. Implementation Details

### C.1. Compute Resources

Graph-GRPO was trained on a single NVIDIA RTX PRO 6000 GPU (96GB VRAM) and 22 CPU cores. Under this setting, the approximate runtime is 20 hours for each molecular task (Protein Docking and Target Property Generation) and 5 hours for synthetic graph generation.

### C.2. Model Architecture

We strictly follow the default model configurations provided by the official implementation of DeFoG (Qin et al., 2025). The backbone is a Graph Transformer that incorporates Relative Random Walk Probabilities (RRWP) as structural encodings. For all tasks, we set the number of attention heads to 8, the node hidden dimension to $d_x = 256$, and the edge hidden dimension to $d_e = 64$. Consistent with DeFoG, the feed-forward network (FFN) dimension is set equal to the node hidden dimension ($d_{ff} = 256$).

The specific configurations vary slightly between synthetic and molecular tasks, as detailed below:

- **Synthetic Tasks (Planar/Tree):** We use a 10-layer model ($L = 10$) and do not use RRWP ($rrwp\_steps = 0$) or global features, following the lightweight setting of the baseline.

- **Molecular Tasks (Protein Docking/PMO):** We use a 12-layer model ($L = 12$) with RRWP steps set to 20 to capture complex chemical structures.

### C.3. Training and Refinement Configurations

**Training.** The policy network is optimized with AdamW (Loshchilov & Hutter, 2019) ($\beta_1 = 0.9$, $\beta_2 = 0.999$) and weight decay $1 \times 10^{-4}$. The learning rate is initialized at $2 \times 10^{-5}$ and decayed by a factor of 0.5 upon reward plateau, with a minimum of $1 \times 10^{-5}$. To accommodate GPU memory constraints, the physical training batch size is adjusted between 5 and 40 depending on the graph size. We correspondingly tune the gradient accumulation steps to maintain a consistent effective batch size of 200 graphs across all experiments. Gradients are clipped to a maximum norm of 5.0.

**GRPO and Partial Trajectory Training.** We use a group size of $K = 60$ for GRPO (Shao et al., 2024). We adopt asymmetric PPO (Schulman et al., 2017) clipping following DAPO (Yu et al., 2025), with $\epsilon_{low} = 0.2$ and $\epsilon_{high} = 0.28$. Advantages are clipped to $[-5, 5]$, and a KL penalty coefficient $\beta = 0.005$ is used.

**Experimental Settings** To ensure reproducibility, we fix random seeds for all experiments.

- **Synthetic Graph Generation:** Evaluations are performed over 5 independent random seeds.

- **Molecular Optimization:** For both Protein Docking and PMO tasks, we use 3 specific seeds (0, 1, and 2).

All reported results represent the mean values averaged over these seeds. Regarding the oracle budget allocation in the refinement strategy, we first utilize 300 oracle calls for de novo generation to initialize the top-M pool.

Regarding the oracle budget allocation in the refinement strategy, we first utilize 300 oracle calls and select the best M molecules initialize the top-M pool. Subsequently, we generate 150 variants for each molecule in the pool until 2,000 oracle calls are consumed, after which we increase the exploration intensity to generate 500 variants per molecule for the remaining 2,000 to 10,000 oracle calls.

**Dynamic Prior Update.** Prior distributions over node types and graph sizes are updated with momentum $\alpha = 0.05$. A buffer of the top-1000 samples is maintained, and the update is triggered when reward improvement exceeds 0.001.

**Refinement.** We allocate the first 300 oracle calls to initialize the candidate pool. Refinement is then performed with a fixed noise level $t_\epsilon = 0.8$ to provide a robust balance between stability and exploration across all benchmarks, and a Top-M pool size is 5.

## D. Additional Visualizations

In this section, we provide additional visualizations to complement the experimental results in the main text. Figure 7 presents randomly sampled molecules generated by Graph-GRPO, showing diverse scaffolds, ring systems, and functional groups. Figure 8 illustrates the intermediate model predictions during the denoising process, and Figure 9 displays the evolution of generated molecules across different training epochs.

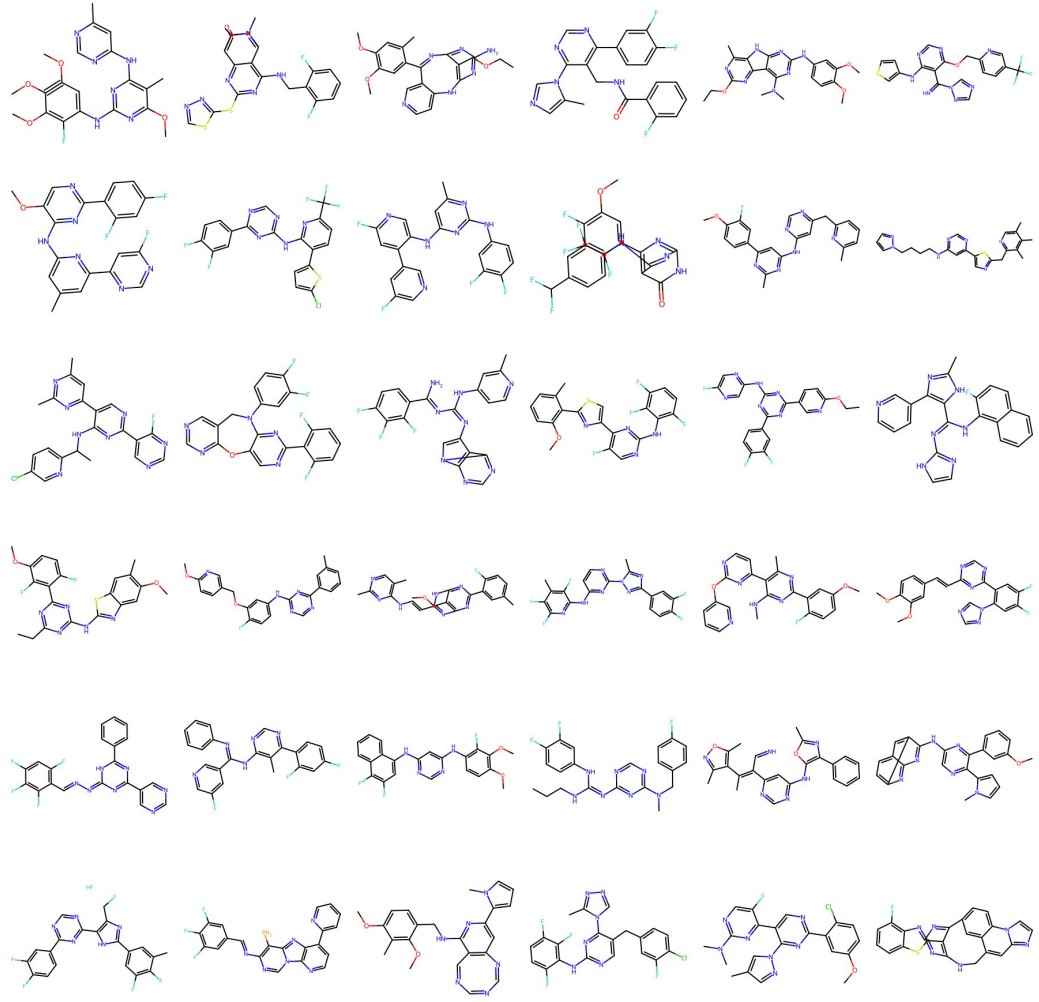

*Figure 7.* Randomly sampled molecules generated by Graph-GRPO. The generated molecules exhibit diverse scaffolds, ring systems, and functional groups, suggesting that the model does not collapse to repeated molecular structures.

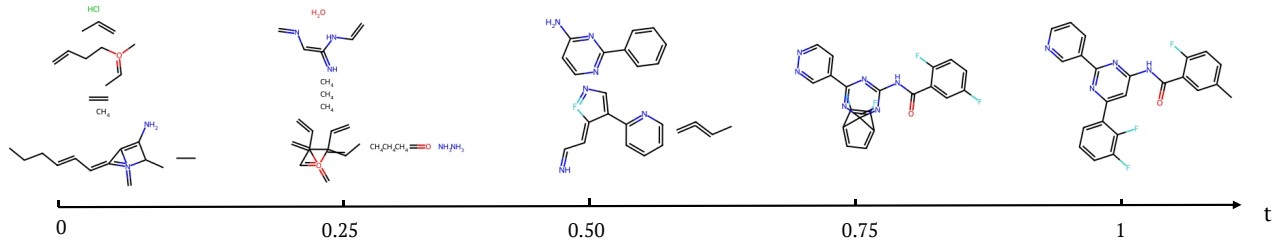

*Figure 8.* **Visualization of predicted clean states $\hat{z}_1$.** We display the clean graphs predicted by the denoiser at different time steps. The model infers the global topology of the target molecule early in the denoising process.

score = 0.02
epoch 0

score = 0.05
epoch 10

score = 0.50
epoch 20

score = 0.39
epoch 30

score = 0.67
epoch 40

score = 0.82
epoch 50

*Figure 9.* **Evolution of generated molecules across training epochs on the JNK3 task.** We showcase samples obtained at different stages of training.

## E. Additional Ablation Studies

In this section, we provide additional ablation studies to further analyze the key design choices and practical efficiency of Graph-GRPO. Specifically, we study the influence of the renoising time $t_\epsilon$ in refinement, compare Graph-GRPO with a REINFORCE variant without analytic transition, and benchmark the computational cost under different graph sizes and denoising steps.

### E.1. Effect of Renoising Time

We evaluate the influence of renoising time $t_\epsilon$, a crucial hyperparameter in the refinement. We vary the value $t_\epsilon$ from 0.0 to 0.9, and report the results in Table 5. It can be observed that the best results are obtained with a large value of $t_\epsilon$, *i.e.*, 0.7 or 0.9. This is because a larger $t_\epsilon$ indicates a smaller noise perturbation for promising candidates, allowing for controlled exploration. In contrast, smaller values of $t_\epsilon$ may destroy molecular structures, leading to performance degeneration, but still perform better than de novo generation ($t_\epsilon = 0.0$).

*Table 5.* Effect of renoising time $t_\epsilon$ on the PMO benchmark.

| Task | 0.9 | 0.7 | 0.5 | 0.3 | 0.0 |
|---|---|---|---|---|---|
| Deco Hop | 0.682 | **0.709** | 0.641 | 0.646 | 0.638 |
| Osimertinib MPO | **0.918** | 0.899 | 0.865 | 0.862 | 0.878 |
| Valsartan SMARTS | **0.949** | 0.827 | 0.584 | 0.211 | 0.174 |

### E.2. Comparison with REINFORCE without Analytic Transition

We further compare Graph-GRPO with a REINFORCE variant that does not use the proposed analytic transition. Both methods are evaluated on the Tree generation task with 64 nodes. They use the same pre-trained DeFoG backbone, identical hyperparameters, including learning rate $1 \times 10^{-5}$, group size 20, 50 denoising steps, and 120 sampled groups per epoch, as well as the same trajectory-level broadcast reward.

The key difference lies in the training objective. Graph-GRPO uses the analytic rate matrix in Proposition 3.1 for both sampling and training, and optimizes the transition probability $\log p(z_{t+\Delta t}|z_t)$. In contrast, the REINFORCE variant samples

trajectories using the original Monte Carlo rate matrix in DeFoG. Since the transition probability is not differentiable without the analytic formulation, REINFORCE instead optimizes the endpoint prediction $\log p_\theta(z_1|z_t)$.

*Table 6.* Comparison with REINFORCE variant on Tree ($N = 64$).

| Method | Reward @Step 0 | Reward @Step 500 | Grad Norm | Clip Rate |
|---|---|---|---|---|
| REINFORCE w/o analytic | 1.03 | 0.312 | 130.6 | 100% |
| **Graph-GRPO w/ analytic** | 1.03 | **1.537** | **0.97** | **0.7%** |

As shown in Table 6, REINFORCE suffers from unstable training and its reward drops from 1.03 to 0.312. By contrast, Graph-GRPO improves the reward to 1.537 with a much smaller gradient norm and a substantially lower gradient clipping rate. This demonstrates that the analytic transition provides a stable and aligned training signal for RL optimization.

Without the analytic transition, the optimized objective $\log p_\theta(z_1|z_t)$ is misaligned with the actual sampling process, which is governed by the one-step transition distribution $p(z_{t+\Delta t}|z_t)$. Since $z_{t+\Delta t}$ is only a small step away from $z_t$ rather than the clean endpoint $z_1$, the resulting gradient signal becomes unstable. The analytic formulation resolves this mismatch by making the training objective exactly match the transition distribution used during sampling.

### E.3. Scalability Analysis

We benchmark the per-graph computational cost of Graph-GRPO on the Tree task using a single GPU. The runtime is averaged over 5 epochs and measured in milliseconds per graph, while memory is measured in GB per graph. We study scalability with respect to both graph size $N$ and trajectory length $T$.

*Table 7.* Scalability analysis of Graph-GRPO computational cost.

*(a)* Scaling with graph size $N$ (fixed $T = 50$).

| $N$ | Sample Time | Train Time | Sample Mem. | Train Mem. |
|---|---|---|---|---|
| 32 | 22.9 | 131.2 | 0.37 | 0.78 |
| 48 | 18.0 | 199.6 | 0.64 | 1.56 |
| 64 | 35.3 | 293.0 | 1.01 | 2.53 |
| 80 | 52.0 | 425.6 | 1.49 | 3.89 |
| 96 | 77.9 | 578.5 | 2.08 | 5.44 |

*(b)* Scaling with denoising steps $T$ (fixed $N = 64$).

| $T$ | Sample Time | Train Time | Sample Mem. | Train Mem. |
|---|---|---|---|---|
| 50 | 35.3 | 293.0 | 1.01 | 2.53 |
| 100 | 60.4 | 525.0 | 1.02 | 4.91 |
| 150 | 90.5 | 780.4 | 1.04 | 7.47 |
| 200 | 118.8 | 1043.9 | 1.05 | 9.84 |

Table 7a reports the cost under different graph sizes with fixed denoising steps $T = 50$. The training memory grows approximately with $O(N^2)$, mainly due to the graph Transformer attention over node-edge structures. For example, increasing $N$ from 32 to 96 leads to a 7.0× increase in training memory, which is close to the theoretical $N^2$ growth.

Table 7b reports the cost under different denoising steps with fixed graph size $N = 64$. Training time and memory scale nearly linearly with $T$. When increasing $T$ from 50 to 200, training time and memory increase by 3.56× and 3.89×, respectively, close to the theoretical 4.0× growth. In contrast, sampling memory remains almost unchanged across different trajectory lengths.

# F. Reward Details

Following previous works and standard evaluation benchmarks, we establish a unified protocol for handling generated graphs with disconnected components. Specifically, for the **Protein Docking** task, we extract the largest connected component (LCC) of the generated graph prior to scoring. For **Target Property Generation** tasks, we directly utilize the standard oracle functions provided by the PMO framework (Gao et al., 2022) to calculate rewards.

## F.1. Protein Docking Rewards

In the protein docking optimization task, we employ a composite reward function consisting of four terms, each strictly bounded in $[0, 1]$. The detailed definitions are as follows:

- **Drug-likeness ($R_{\text{QED}}$):** This component ensures basic drug-like properties, defined as an indicator function $\mathbb{I}(\text{QED}(G) > 0.5)$.

- **Synthetic Accessibility ($R_{\text{SA}}$):** This represents the normalized synthetic accessibility score, calculated as $(10 - \text{SA}(G))/9$. It favors molecules that are easier to synthesize (lower SA values).

- **Novelty ($R_{\text{Nov}}$):** This term encourages exploration beyond the training distribution $\mathcal{D}_{\text{train}}$. It is computed as $1 - \max_{G' \in \mathcal{D}_{\text{train}}} \text{Sim}(G, G')$, where Sim denotes the Tanimoto similarity of Morgan fingerprints.

- **Docking Score ($R_{\text{DS}}$):** This is the normalized docking score derived from the binding energy $E$ (kcal/mol), formulated as $R_{\text{DS}} = \text{Clip}(-E/20, 0, 1)$, where lower energy corresponds to a higher reward.

**Early Pruning.** To improve training efficiency, we also implement the early-pruning strategy from GDPO (Liu et al., 2024): the computationally expensive docking simulation ($R_{\text{DS}}$) is performed only for molecules that satisfy the drug-likeness constraint ($R_{\text{QED}} > 0$); otherwise, $R_{\text{DS}}$ is set to 0.

## F.2. Target Property Optimization Rewards

For the *Valsartan SMARTS* task, the objective is to generate molecules containing a specific substructure while matching the physicochemical properties of a reference molecule (Sitagliptin). The total reward $R(G)$ is defined as the geometric mean of a binary substructure score $s_{\text{struct}} \in \{0, 1\}$ and three physicochemical property scores $\{s_1, s_2, s_3\} \in (0, 1]$:

$$R(G) = (s_{\text{struct}} \cdot s_1 \cdot s_2 \cdot s_3)^{\frac{1}{4}}. \tag{33}$$

The property scores $s_i$ are calculated using Gaussian kernels centered at the reference values, which decay exponentially as the molecule's properties deviate from the target.

**The Sparse Reward Challenge.** This geometric mean formulation creates a severe optimization bottleneck during the cold-start phase. Even in the rare event that the model accidentally generates a valid scaffold ($s_{\text{struct}} = 1$), the associated physicochemical properties typically deviate significantly from the specific targets of Sitagliptin. Due to the rapid decay of Gaussian kernels, the property scores $s_i$ drop to near-zero values. Consequently, the multiplicative nature of the geometric mean suppresses the total reward $R(G)$ to a negligible magnitude (e.g., $< 10^{-4}$). This results in an extremely weak gradient signal that is easily drowned out by training noise, preventing the model from reinforcing these valid structural discoveries.

**Solution: Two-Stage Training Curriculum.** To address the vanishing gradient issue, we implement a curriculum strategy. We emphasize that this is strictly a training-time scheduling technique designed to bootstrap the policy, and is distinct from the refinement strategy used during inference. The training process is divided into two phases:

1. **Training Stage 1 (Structural Warm-up):** We first optimize the policy using a simplified objective that focuses solely on the binary substructure constraint ($R = s_{\text{struct}}$). Unlike the geometric mean, this provides a clear, non-vanishing gradient signal, enabling the policy to quickly learn the topological requirements of the target scaffold.

2. **Training Stage 2 (Multi-Objective Training):** We use the checkpoint saved from Stage 1 to initialize the full RL training. With the policy already capable of generating the correct scaffold, the geometric mean reward becomes non-zero ($R > 0$). This allows the RL algorithm to effectively optimize the remaining physicochemical property scores.

# G. Additional Results

## G.1. Protein Docking Metrics.

Table 8 details the generative metrics for the five target proteins. The results show that Graph-GRPO maintains near-perfect validity and uniqueness (100%) while optimizing for high binding affinities.

*Table 8.* Detailed performance metrics of Graph-GRPO on five distinct targets.

| Target | Valid. | Uniq. | Nov. | Avg. QED | Avg. SA | Avg. Reward |
|--------|--------|-------|------|----------|---------|-------------|
| *parp1* | 97.803 | 100.000 | 0.945 | 0.667 | 0.655 | 0.459 |
| *fa7* | 98.617 | 100.000 | 0.992 | 0.726 | 0.650 | 0.358 |
| *braf* | 98.405 | 100.000 | 0.976 | 0.723 | 0.679 | 0.447 |
| *jak2* | 98.763 | 100.000 | 0.987 | 0.750 | 0.674 | 0.466 |
| *5ht1b* | 98.145 | 100.000 | 0.977 | 0.720 | 0.675 | 0.421 |

## G.2. Synthetic Graph Generation.

Table 9 presents the extended comparison on the Planar and Tree datasets. For fair and consistent comparison, **all baseline results (including DeFoG) are directly cited from the original DeFoG paper (Qin et al., 2025)**. We strictly followed their evaluation protocol to evaluate Graph-GRPO under the same settings.

*Table 9.* **Extended Graph Generation Performance on Synthetic Graphs.** We report the detailed MMD metrics and validity statistics. We only report the performance of Graph-GRPO (Ours) based on our own experiments, averaged over 5 runs.

| | Planar Dataset | | | | | | | | | |
|-------|------|------|------|------|------|------|------|------|------|------|
| **Model** | Deg. ↓ | Clus. ↓ | Orbit ↓ | Spec. ↓ | Wavelet ↓ | Ratio ↓ | Valid ↑ | Unique ↑ | Novel ↑ | V.U.N. ↑ |
| Train set | 0.0002 | 0.0310 | 0.0005 | 0.0038 | 0.0012 | 1.0 | 100 | 100 | 0.0 | 0.0 |
| GraphRNN | 0.0049 | 0.2779 | 1.2543 | 0.0459 | 0.1034 | 490.2 | 0.0 | 100 | 100 | 0.0 |
| GRAN | 0.0007 | 0.0426 | 0.0009 | 0.0075 | 0.0019 | 2.0 | 97.5 | 85.0 | 2.5 | 0.0 |
| SPECTRE | 0.0005 | 0.0785 | 0.0012 | 0.0112 | 0.0059 | 3.0 | 25.0 | 100 | 100 | 25.0 |
| DiGress | 0.0007 | 0.0780 | 0.0079 | 0.0098 | 0.0031 | 5.1 | 77.5 | 100 | 100 | 77.5 |
| EDGE | 0.0761 | 0.3229 | 0.7737 | 0.0957 | 0.3627 | 431.4 | 0.0 | 100 | 100 | 0.0 |
| BwR | 0.0231 | 0.2596 | 0.5473 | 0.0444 | 0.1314 | 251.9 | 0.0 | 100 | 100 | 0.0 |
| BiGG | 0.0007 | 0.0570 | 0.0367 | 0.0105 | 0.0052 | 16.0 | 62.5 | 85.0 | 42.5 | 5.0 |
| GraphGen | 0.0328 | 0.2106 | 0.4236 | 0.0430 | 0.0989 | 210.3 | 7.5 | 100 | 100 | 7.5 |
| HSpectre (one-shot) | 0.0003 | 0.0245 | 0.0006 | 0.0104 | 0.0030 | 1.7 | 67.5 | 100 | 100 | 67.5 |
| HSpectre | 0.0005 | 0.0626 | 0.0017 | 0.0075 | 0.0013 | 2.1 | 95.0 | 100 | 100 | 95.0 |
| GruM | 0.0004 | 0.0301 | 0.0002 | 0.0104 | 0.0020 | 1.8 | — | — | — | 90.0 |
| CatFlow | 0.0003 | 0.0403 | 0.0008 | — | — | — | — | — | — | 80.0 |
| DisCo | 0.0002 | 0.0403 | 0.0009 | — | — | — | 83.6 | 100.0 | 100.0 | 83.6 |
| Cometh - PC | 0.0006 | 0.0434 | 0.0016 | 0.0049 | — | — | 99.5 | 100.0 | 100.0 | 99.5 |
| DeFoG | 0.0005 | 0.0501 | 0.0006 | 0.0072 | 0.0014 | 1.6 | 99.5 | 100.0 | 100.0 | 99.5 |
| **Graph-GRPO (50 steps)** | 0.0002 | 0.0325 | 0.0012 | 0.0067 | 0.0013 | 1.5 | 95.0 | 100.0 | 100.0 | 95.0 |

| | Tree Dataset | | | | | | | | | |
|-------|------|------|------|------|------|------|------|------|------|------|
| **Model** | Deg. ↓ | Clus. ↓ | Orbit ↓ | Spec. ↓ | Wavelet ↓ | Ratio ↓ | Valid ↑ | Unique ↑ | Novel ↑ | V.U.N. ↑ |
| Train set | 0.0001 | 0.0000 | 0.0000 | 0.0075 | 0.0030 | 1.0 | 100 | 100 | 0.0 | 0.0 |
| GRAN | 0.1884 | 0.0080 | 0.0199 | 0.2751 | 0.3274 | 607.0 | 0.0 | 100 | 100 | 0.0 |
| DiGress | 0.0002 | 0.0000 | 0.0000 | 0.0113 | 0.0043 | 1.6 | 90.0 | 100 | 100 | 90.0 |
| EDGE | 0.2678 | 0.0000 | 0.7357 | 0.2247 | 0.4230 | 850.7 | 0.0 | 7.5 | 100 | 0.0 |
| BwR | 0.0016 | 0.1239 | 0.0003 | 0.0480 | 0.0388 | 11.4 | 0.0 | 100 | 100 | 0.0 |
| BiGG | 0.0014 | 0.0000 | 0.0000 | 0.0119 | 0.0058 | 5.2 | 100 | 87.5 | 50.0 | 75.0 |
| GraphGen | 0.0105 | 0.0000 | 0.0000 | 0.0153 | 0.0122 | 33.2 | 95.0 | 100 | 100 | 95.0 |
| HSpectre (one-shot) | 0.0004 | 0.0000 | 0.0000 | 0.0080 | 0.0055 | 2.1 | 82.5 | 100 | 100 | 82.5 |
| HSpectre | 0.0001 | 0.0000 | 0.0000 | 0.0117 | 0.0047 | 4.0 | 100 | 100 | 100 | 100 |
| DeFoG | 0.0002 | 0.0000 | 0.0000 | 0.0108 | 0.0046 | 1.6 | 96.5 | 100.0 | 100.0 | 96.5 |
| **Graph-GRPO (50 steps)** | 0.0004 | 0.0000 | 0.0000 | 0.0105 | 0.0045 | 2.2 | 97.5 | 100.0 | 100.0 | 97.5 |

