# OpenReview forum: "Graph-GRPO: Training Graph Flow Models with Reinforcement Learning"
_ICML.cc/2026/Conference — ICML 2026 regular_

### Official Review · Reviewer_7w6u · 2026-03-09

**Soundness:** 4
**Presentation:** 3
**Significance:** 3
**Originality:** 2
**Overall Recommendation:** 4
**Confidence:** 4

**Summary:**

The paper presents an adaption of GRPO, a popular reinforcement learning framework, to graph discrete diffusion models. The authors derive a differentiable expression for the CTMC rate matrix, allowing the stable fine-tuning of a pretrained graph discrete diffusion model. They also explore a refinement mechanism that consists in renoising promising samples, and regenerating them.

**Compliance With Llm Reviewing Policy:**

Affirmed.

**Final Justification:**

Though I maintain that the authors' contribution is slightly incremental, the framework that the authors have put together obtain a strong performance on the considered benchmarks. Most of my minor concerns and questions have been answered by the reviewers.

Given that the topic of RL applied to graph diffusion models remains mostly underexplored, I believe that Graph-GRPO is a valuable contribution for the graph diffusion literature. I therefore increased my score from 3 to 4.

**Key Questions For Authors:**

- On Planar and Tree, you report results using 50 steps. Does that include additional steps induced by the renoising-regeneration process ?

- In the reward used for synthetic graphs (eq. 17), what is S_k ?

**Limitations:**

yes

**Strengths And Weaknesses:**

**Soundness** : The paper is theoretically sound. The proofs appear to be correct, and the expression derived in Proposition 3.1. allows differentiation through the rate matrix, which seems to be a nice contribution. I am less familiar with the RL component of this work, which limits my ability to thoroughly assess the technical soundness and design choices regarding GRPO.

**Presentation** : The paper is overall well-presented, up to a major drawback : while discrete diffusion models are extensively described in the Section 2 (Preliminary), there is no introduction to reinforcement learning and GRPO. The paper dives into technical considerations without introducing the topic. For instance, Section 3.1 l.160 refers to "the old and new policy models", even though these were neither mentioned nor introduced beforehand. Nitpick : DeFoG (Qin et al.) and Cometh (Siraudin et al.) use RRWP, not RWSE.

**Significance** : though the method seems strong on the evaluated molecular benchmarks (protein docking and molecular optimization), I feel like most of the focus on molecular generative models has shifted towards 3D models, especially regarding protein docking. In my opinion, this weakens the significance of the experimental section.

**Originality** : I feel that this is the main weakness of the paper. Apart from Prop. 3.1, the work seems to be a straightforward adaptation of GRPO. The idea of renoising and regenerating is not particularly new either (see [1], [2]). Consequently, I think that the contribution of this work is incremental.

[1] Fast T2T: Optimization Consistency Speeds Up Diffusion-Based Training-to-Testing Solving for Combinatorial Optimization, Li et al, NeurIPS 2024.
[2] Restart Sampling for Improving Generative Processes, Xu et al., NeurIPS 2023

---

> ### Author Rebuttal · Authors · 2026-03-30
>
> We appreciate the reviewer for the careful reading and for identifying the missing RL/GRPO introduction.
>
> > **Originality.** Apart from Proposition 3.1, is this an incremental adaptation of GRPO?
>
> **A1: We respectfully disagree.** Our contribution is a complete RL framework for discrete graph flow models, not an adaptation of a specific RL algorithm. GRPO is simply the policy optimization method we use for training; the framework contributions beyond Proposition 3.1 include: (1) the refinement strategy with reward-ranked priority pool for inference-time exploration (Section 3.3), (2) the dynamic prior that adapts the noise distribution during training (Appendix B), and (3) the overall pipeline design for formulating discrete graph generation as a tractable RL problem (trajectory rollout, advantage computation, training-inference alignment). Proposition 3.1 is the key enabler that makes this framework possible, but the framework itself is broader than any single component.
>
> > **Significance.** The broader field focuses on 3D molecular models; how significant are 2D results?
>
> **A2: 3D molecular generation is indeed an important direction.** However, the focus of this paper is the RL optimization framework for graph flow models, not the molecular representation itself.
>
> To validate the quality of generated graphs, we follow InVirtuoGen [1] and GenMol [2] by evaluating on the widely adopted PMO benchmark and protein docking tasks. Notably, these baselines operate on textual SMILES representations, which lack explicit structural information. Our graph-based approach outperforms them on the same benchmarks (Table 3), demonstrating that richer structural features do improve generation quality.
>
> These experiments serve as validation of our optimization framework, which is model-agnostic and may apply equally to 3D graph flow models as they become available.
>
> > **Presentation.** Missing RL/GRPO introduction; DeFoG and Cometh use RRWP, not RWSE.
>
> A3: We thank the reviewer for this suggestion. We will add a dedicated subsection introducing GRPO and the old/new policy distinction before Section 3.1. We will also correct the notation: DeFoG and Cometh use RRWP (Relative Random Walk Probabilities), not RWSE. Both will appear in the revision.
>
> > **Q1:** Do the 50 steps reported on Planar and Tree include the additional steps induced by the renoising-regeneration process?
>
> **A4: Planar and Tree experiments do not use the refinement mechanism. The 50 steps are pure flow matching sampling steps.** For molecular optimization tasks (PMO, docking) where refinement is used, the oracle calls consumed by renoising-regeneration steps are fully counted toward the total oracle budget.
>
> > **Q2:** In the reward used for synthetic graphs (Eq. 17), what is $S_k$?
>
> **A5:** $S_k$ denotes empirical statistics of the training set graphs: degree distribution, clustering coefficients, and orbit counts. The reward measures how well the generated graph matches these training distribution statistics. We will add explicit notation in the revision.
>
> > **Renoising-regenerating novelty.** Similar ideas exist in prior work.
>
> **A6: The general principle of renoising-regeneration exists in prior work** such as Restart Sampling [3] and Fast T2T [4]. The key difference in our refinement is **reward-guided selective exploration**: rather than renoising all samples, we maintain a reward-ranked priority pool $\mathcal{B}$ (Section 3.3) and selectively refine only the top-M high-reward candidates. One goal of this design is to conserve oracle calls: in molecular optimization each oracle evaluation is expensive, so selectively refining high-reward candidates rather than exploring blindly improves generation efficiency under a limited oracle budget.
>
> Our refinement also operates in discrete graph space via the conditional probability path (Eq. 16) with a controllable perturbation level $t_e$ (Table 5), whereas Restart Sampling operates in continuous image space and Fast T2T uses consistency models for combinatorial optimization. We will add this discussion and cite both papers in the revision.
>
> **References:**
>
> [1] Kaech et al., "InVirtuoGen: Refine drugs, don't complete them", CoRR 2025.
>
> [2] Lee et al., "GenMol: A drug discovery generalist with discrete diffusion", ICML 2025.
>
> [3] Xu et al., "Restart sampling for improving generative processes", NeurIPS 2023.
>
> [4] Li et al., "Fast T2T: Optimization consistency speeds up diffusion-based training-to-testing solving", NeurIPS 2024.

---

> > ### Author Rebuttal · Reviewer_7w6u · 2026-04-03
> >
> > Thank you for writing a detailed rebuttal.
> >
> > Overall, I think that the authors did a good job answering my questions and concerns. I still feel that the contribution is slightly incremental though. The central part of the model is the GRPO algorithm, which the authors have adapted to discrete diffusion.
> >
> > Neverthess I acknowledge that the framework that the authors have put together achieves strong empirical results. I will increase my score to 4.

---

> > > ### Author Response · Authors · 2026-04-04
> > >
> > > Dear Reviewer 7w6u,
> > >
> > > We sincerely appreciate your helpful suggestions and are glad to know that our revision has addressed most of your concerns.
> > >
> > > Since your current comments appear positive, we would be truly grateful if you could consider updating the Overall Recommendation.
> > >
> > > Thank you again for your time and support.

---

### Official Review · Reviewer_zAF2 · 2026-03-10

**Soundness:** 3
**Presentation:** 3
**Significance:** 2
**Originality:** 3
**Overall Recommendation:** 3
**Confidence:** 3

**Summary:**

This paper proposes Graph-GRPO, an RL framework for graph flow models under verifiable rewards. The main technical claim is that prior GFM sampling relies on Monte Carlo transition estimation, which is poorly suited for policy-gradient optimization, and that this can be addressed by deriving an analytic transition probability. The method is combined with a refinement strategy that perturbs and regenerates promising samples. Experiments are reported on synthetic graph generation, protein docking, and PMO molecular optimization, where the method improves over the DeFoG backbone and several baselines.

**Compliance With Llm Reviewing Policy:**

Affirmed.

**Final Justification:**

The rebuttal of the authors has resolved my concerns.  I keep my original score at last.

**Key Questions For Authors:**

1.Could the authors provide a more direct same-backbone comparison, e.g., RL without the analytic transition or with an alternative surrogate transition relaxation?
2.To what extent do the authors expect the proposed approach to generalize beyond molecular graphs to other graph-generation settings with stronger structural dependencies? The paper is motivated in relatively general terms, but most experiments focus on molecular optimization. A clearer discussion of the expected scope of applicability would help calibrate the contribution more appropriately.
3.Can the authors clarify under what assumptions the analytic transition formulation is exact, and when it should instead be viewed as an approximation?

**Limitations:**

Yes.

**Strengths And Weaknesses:**

Strengths：
1.This paper addresses an important problem: how to align graph generative models with downstream reward objectives in a principled way which is a meaningful direction.
2.The proposed analytic transition formulation is interesting. It targets a real limitation of prior GFM-based sampling, namely that Monte Carlo transition estimation is not well suited for stable RL optimization.
3.The empirical section is fairly broad. The paper evaluates the method on synthetic graph tasks, docking, and PMO optimization.
Weaknesses:
1.The analytic transition derivation appears to depend on fairly restrictive modeling assumptions, but the paper does not sufficiently clarify the exact scope of these assumptions, nor does it clearly separate what is exact from what is effectively an approximation.
2.The empirical gains are not cleanly attributable to the claimed contribution. The final system includes multiple additional ingredients beyond the analytic transition idea itself, including refinement, prescreening, dynamic priors, and buffer-based mechanisms.The current ablations are not sufficient to resolve this.

---

> ### Author Rebuttal · Authors · 2026-03-30
>
> We thank the reviewer for the detailed critique and address each concern below.
>
> > **W2 / Q1: Can you provide a direct comparison, e.g., RL without the analytic transition?**
>
> **A1:** We provide a direct same-backbone ablation on Tree generation (64 nodes): Graph-GRPO (with analytic transition) vs. REINFORCE (without analytic transition).
>
> Experiment setup. Both methods use the same pre-trained DeFoG backbone, identical hyperparameters (lr=1e-5, group_size=20, 50 denoising steps, 120 sample groups per epoch), and the same trajectory-level broadcast reward. The difference is:
>
> - **Graph-GRPO**: identical to the paper, uses the analytic rate matrix (Prop. 3.1) for both sampling and training, optimizing $\log p(z_{t+\Delta t}|z_t)$.
> - **REINFORCE (w/o analytic)**: uses DeFoG's original MC-based rate matrix for sampling. Since no differentiable transition probability is available without the analytic formula, training uses REINFORCE  to optimize $\log p_\theta(z_1|z_t)$, the model's endpoint prediction.
>
> Results (Tree, 64 nodes):
>
> | Method | Reward @epoch 0 | Reward @epoch 500 | Training Grad Norm | Grad Clip Rate |
> |---|---|---|---|---|
> | REINFORCE (RL w/o analytic) | 1.03 | 0.312 | 130.6 | 100% |
> | Graph-GRPO (RL w/ analytic) | 1.03 | **1.537** | **0.97** | **0.7%** |
>
> REINFORCE's reward drops from 1.03 to 0.312 (−70%), while Graph-GRPO rises to 1.537 (+49%). The analytic transition provides stable gradient signals (training grad norm 0.97 vs. 130.6), enabling a stable training process.
>
> In discrete flow matching, without the analytic transition, training optimizes $\log p_\theta(z_1|z_t)$ (the endpoint prediction), but sampling is governed by $p(z_{t+\Delta t}|z_t)$, a single-step transition. Since $z_{t+\Delta t}$ is only a small step away from $z_t$ rather than the endpoint $z_1$, the gradient signal is misaligned with the actual sampling process, causing training to collapse.
>
> The analytic formulation (Prop. 3.1) resolves this by making the training objective $\log p(z_{t+\Delta t}|z_t)$ exactly the distribution used during sampling, ensuring alignment between optimization and generation. We conduct this ablation on Tree because its fast reward computation enables rapid iteration; the conclusion generalizes since the gradient instability is inherent to the training objective mismatch, independent of the downstream task.
>
> For the contributions of other components (refinement, prescreening, dynamic prior), we provide a full ablation table in Reviewer vXSd A1.
>
> > **W1 / Q3: Assumption scope.** Under what assumptions is the analytic transition exact, and when is it an approximation?
>
> **A2: All our assumptions are inherited from prior discrete flow matching methods.** We replace one existing approximation (MC sampling) with an exact computation. The shared assumptions are:
>
> 1. Node-edge factorization (Eq. 1): $p(G) = \prod_i p(x^i) \cdot \prod_j p(e^j)$, reducing the state space from $|\mathcal{X}|^N \times |\mathcal{E}|^{N^2}$ to $N \times |\mathcal{X}| + N^2 \times |\mathcal{E}|$.
> 2. Euler discretization: $p(z_{t+\Delta t}|z_t) = \delta(z_t, z_{t+\Delta t}) + R_t^\theta(z_t, z_{t+\Delta t}) \cdot \Delta t$, a first-order approximation exact as $\Delta t \to 0$, standard in continuous-time discrete flow matching [1].
> 3. Full support of $p_0$, ensuring $Z_t^{>0} = S$ for all $t < 1$. This is a standard assumption for well-defined rate matrices.
>
> These are standard and well-established in the discrete flow matching literature, adopted by prior work including [1] and DeFoG [2].
>
> > **Q2: Generalization.** Can this approach generalize beyond molecular graphs?
>
> **A3: Graph-GRPO is model-agnostic and task-agnostic** within the family of discrete graph flow models. We validate this generality on both synthetic graph generation (Planar, Tree) and molecular tasks (protein docking, PMO).
>
> The current focus on relatively small-scale graphs is primarily constrained by GPU memory, computational cost, and the sampling efficiency of existing graph flow models, not by the RL framework itself. Molecular generation is a representative and practically important application within this scale.
>
> As hardware and model sampling efficiency improve, Graph-GRPO can be directly applied to larger graphs without modification.
>
> **References:**
>
> [1] Campbell et al., "A continuous time framework for discrete denoising models", NeurIPS 2022.
>
> [2] Qin et al., "DeFoG: Discrete flow matching for graph generation", ICML 2025.

---

> > ### Author Rebuttal · Reviewer_zAF2 · 2026-04-03
> >
> > My question has been answered, and I have no further questions.

---

> > > ### Author Response · Authors · 2026-04-04
> > >
> > > Dear Reviewer zAF2,
> > >
> > > Thank you for the positive acknowledgement. We are glad the new experiments and clarifications addressed your concerns. If you feel the updates warrant it, we would appreciate your reconsidering the score. Thank you again for the valuable feedback.

---

### Official Review · Reviewer_jMcw · 2026-03-12

**Soundness:** 3
**Presentation:** 3
**Significance:** 3
**Originality:** 3
**Overall Recommendation:** 4
**Confidence:** 3

**Summary:**

This paper studies reinforcement learning for graph generative models. The authors propose Graph-GRPO, a framework that enables policy gradient optimization for graph flow networks (GFNs). The key technical contribution is an analytical formulation of the transition rate in graph flow models, replacing the Monte Carlo estimation used in prior work and allowing differentiable trajectory likelihoods. Based on this formulation, the authors apply GRPO to optimize the generative policy. The paper further introduces a refinement strategy that perturbs generated graphs to perform localized exploration during training. The method is evaluated on synthetic graph generation benchmarks and molecular optimization tasks (docking and PMO). Experimental results show improvements in reward-based metrics and validity-related measures compared with several baseline methods.

**Compliance With Llm Reviewing Policy:**

Affirmed.

**Final Justification:**

The authors' response has addressed my issue. I would like to maintain my positive recommendation.

**Key Questions For Authors:**

Q1. Could the authors clarify whether Eq. (10) is intended to define the full rate matrix or only the off-diagonal entries?
Q2. Does the analytical transition formulation produce transition dynamics that are theoretically equivalent to the Monte Carlo estimator used in prior graph flow models, or should it be interpreted as an expectation-level approximation?
Q3. Could the authors include an ablation comparing reinforcement learning using the original Monte Carlo transition estimator versus reinforcement learning using the proposed analytical formulation?
Q4. The PMO system includes refinement, prescreening, adaptive priors, and other components. Could the authors quantify the individual contributions of these components relative to the analytical transition formulation?
Q5. How does the training complexity of Graph-GRPO scale with graph size and trajectory length? Providing more detailed comparisons of runtime or sample efficiency would strengthen the practical evaluation.

**Limitations:**

Yes.

**Strengths And Weaknesses:**

S1. Differentiable transition formulation for graph flow models.
The analytical transition-rate formulation is a technically meaningful contribution. By removing the need for Monte Carlo estimation, the proposed formulation allows reinforcement learning objectives to be directly applied to graph flow generative models. This addresses a real limitation of existing approaches.
S2. Reinforcement learning formulation aligned with trajectory likelihoods.
The framework provides a coherent way to integrate reinforcement learning optimization with graph flow models. The formulation aligns trajectory probabilities with policy gradient training, which is conceptually consistent with generative modeling objectives.
S3. Empirical evaluation on both synthetic and domain-specific tasks.
The experiments include both synthetic graph generation benchmarks and molecular optimization tasks, providing evidence that the approach can improve reward-oriented objectives in different settings.
S4. Practical exploration mechanism.
The refinement strategy provides a simple mechanism for improving exploration during training and appears to help mitigate sparse reward issues in graph generation.
W1. The theoretical scope of the analytical transition formulation is not fully clarified.
The derivation in the appendix appears to justify the closed-form rate expression mainly for off-diagonal transitions and within the active-flow regime, while the main text presents Eq. (10) as defining the full transition rate matrix. Clarifying this distinction would strengthen the technical soundness of the formulation.
W2. The relationship between the analytical transition formulation and the original Monte Carlo estimator is unclear.
The paper replaces Monte Carlo transition estimation with an analytical expectation, but it is not clear whether the resulting dynamics are theoretically equivalent to the original estimator or represent an approximation.
W3. The experiments do not isolate the contribution of the core methodological idea.
The strongest results rely on a system that includes refinement, prescreening, adaptive priors, and other components, making it difficult to attribute the observed gains specifically to the analytical transition formulation.
W4. Limited analysis of efficiency and scalability.
The paper provides little discussion of computational cost, training efficiency, or scalability relative to existing methods.
W5. Experimental analysis is somewhat limited.
The evaluation focuses mainly on final reward metrics, with limited diagnostic analysis of training dynamics, sample efficiency, or diversity of generated graphs.

---

> ### Author Rebuttal · Authors · 2026-03-30
>
> We thank the reviewer for the thorough and technically precise critique.
>
> > **W1 / Q1: Theoretical scope.** Does Eq. (10) define the full rate matrix or only off-diagonal entries?
>
> **A1: Eq. (10) defines only the off-diagonal entries** ($z_{t+\Delta t} \neq z_t$). We compute transition probabilities to other categories via $p(z_{t+\Delta t}|z_t) = R_t^\theta(z_t, z_{t+\Delta t}) \cdot \Delta t$, and the staying probability as $p(z_{t+\Delta t} = z_t | z_t) = 1 - \sum_{z_{t+\Delta t} \neq z_t} p(z_{t+\Delta t} | z_t)$ (Eq. 29, Appendix A.3), ensuring a valid probability distribution. We will add a note in the main text.
>
> > **W2 / Q2: Analytic vs. Monte Carlo sampling.** Are they theoretically equivalent, or is one an approximation?
>
> **A2: The MC estimator is an approximation of our analytic result, not the other way around.**
>
> As shown in Eq. (9), the unconditional rate matrix is the expectation:
>
> $$R\_t(z\_t, z\_{t+\Delta t}) = \mathbb{E}\_{\hat{z}\_1 \sim p\_\theta(\cdot|z\_t)} [R\_t(z\_t, z\_{t+\Delta t}|\hat{z}\_1)]$$
>
> - **MC estimator** (Algorithm 1): $R_t^{MC} = R_t(z_t, z_{t+\Delta t}|\hat{z}_1)$, where $\hat{z}\_1 \sim p\_\theta(\cdot|z\_t)$ is a single sample. This is a stochastic approximation of the expectation.
> - **Our analytic formula** (Proposition 3.1, Eq. 10): $R_t^\theta = \sum_{z_1} p_\theta(z_1|z_t) \cdot R_t(z_t, z_{t+\Delta t}|z_1)$, which Eq. 10 simplifies into a closed-form expression directly computable from $p_\theta$ and $p_0$, without sampling.
>
> > **Q3: Analytic vs. MC ablation.** Can you provide an ablation comparing RL with the original MC estimator vs. the proposed analytical formulation?
>
> **A3: Without the analytic formula, MC sampling breaks gradient flow, making RL training infeasible.** Since no differentiable transition probability is available, we use REINFORCE [1] to optimize $\log p_\theta(z_1|z_t)$ under identical hyperparameters as an ablation. However, this training objective (endpoint prediction) is misaligned with the actual sampling distribution $p(z_{t+\Delta t}|z_t)$, causing training collapse.
>
> On Tree (64 nodes), REINFORCE collapses to 0.312 reward (-70%) while Graph-GRPO reaches 1.537 (+49%). Detailed setup and analysis in our response to Reviewer zAF2 A1.
>
>
>
> > **W3 / Q4: Component isolation.** Can you quantify individual contributions of refinement, prescreening, and dynamic prior?
>
> **A4: RL is the dominant contributor** (+5.82 over DeFoG). Refinement adds +4.17 without RL, +1.05 to +1.54 with RL. Prescreening adds +0.64 without RL, +0.08 to +0.28 with RL. Dynamic prior contributes +0.55 to +1.24. Full table in Reviewer vXSd A1.
>
> > **W4 / Q5: Efficiency and scalability.** How does training complexity scale with graph size and trajectory length?
>
> **A5: We benchmark per-graph training cost on the Tree task** (single GPU, 5-epoch average; time in ms/graph, memory in GB/graph).
>
> Scaling with graph size N (fixed T=50):
>
> |N|Sample|Train|Sample Mem|Train Mem|
> |---|---|---|---|---|
> |32|22.9|131.2|0.37|0.78|
> |48|18.0|199.6|0.64|1.56|
> |64|35.3|293.0|1.01|2.53|
> |80|52.0|425.6|1.49|3.89|
> |96|77.9|578.5|2.08|5.44|
>
> Training memory scales as O(N²) (0.78 GB at N=32 → 5.44 GB at N=96, 7.0× growth vs. N² ratio 9.0×), dominated by GraphTransformer attention.
>
> Scaling with trajectory length T (fixed N=64):
>
> |T|Sample|Train|Sample Mem|Train Mem|
> |---|---|---|---|---|
> |50|35.3|293.0|1.01|2.53|
> |100|60.4|525.0|1.02|4.91|
> |150|90.5|780.4|1.04|7.47|
> |200|118.8|1043.9|1.05|9.84|
>
> Training time and memory scale linearly with T (3.56× and 3.89× at T=200 vs. theoretical 4.0×). Sampling memory remains constant across T. We will include scaling curves in the revision.
>
> > **W5: Training diagnostics.** Lacking diagnostics on training dynamics and sample diversity.
>
> **A6: Training diagnostics on Deco Hop.**
>
> https://imgur.com/x1KiWCk
>
> Average reward increases steadily while policy entropy remains non-degenerate, indicating no mode collapse. The max_reward curve shows the model increasingly samples high-scoring molecules from a narrow high-reward region of the chemical space. Gradient norms stay bounded after clipping, and both losses converge smoothly.
>
> Sample diversity. Following GenMol [2] and DeFoG [3], we visualize 36 molecules randomly sampled without filtering from Deco Hop inference:
>
> https://imgur.com/zOMPRnG
>
> The molecules exhibit diverse scaffolds, ring systems, and functional groups with no repeated structures, confirming no mode collapse.
>
> We will add additional explanations and experiments in the camera-ready paper.
>
> **References:**
>
> [1] Williams, "Simple statistical gradient-following algorithms for connectionist reinforcement learning", Machine Learning 1992.
>
> [2] Lee et al., "GenMol: A drug discovery generalist with discrete diffusion", ICML 2025.
>
> [3] Qin et al., "DeFoG: Discrete flow matching for graph generation", ICML 2025.

---

> > ### Author Rebuttal · Reviewer_jMcw · 2026-04-02
> >
> > Thank you for the clarification. I am satisfied with the response. I would like to maintain my positive recommendation.

---

### Official Review · Reviewer_vXSd · 2026-03-13

**Soundness:** 3
**Presentation:** 4
**Significance:** 2
**Originality:** 3
**Overall Recommendation:** 5
**Confidence:** 2

**Summary:**

Paper proposes an RL framework for graph flow models that replaces Monte Carlo transition estimation with an analytic transition probability and also adds a refinement procedure. The method is evaluated on synthetic graph generation, protein docking, and PMO molecular optimization, where it reports strong empirical gains over several baselines

**Compliance With Llm Reviewing Policy:**

Affirmed.

**Final Justification:**

The authors conducted additional ablation experiments and resolved the questions. The final version should be updated.

**Key Questions For Authors:**

How much of the final gain comes from the RL objective itself, and how much comes from the refinement loop as a search heuristic? A stronger ablation study on equal oracle budgets would help clarify this.
What guarantees or constraints ensure that the analytic transition probabilities remain numerically stable across all time steps and graph states? Please also clarify whether the factorized node or edge modeling assumption limits applicability on graphs with stronger structural dependencies.

**Limitations:**

yes

**Strengths And Weaknesses:**

The main strength is that the paper addresses a real limitation of applying policy-gradient RL to graph flow models and offers a concrete technical modification with clear practical motivation. The empirical section is well explained and broad and shows improvements on multiple benchmarks, it includes difficult molecular optimization settings.
The main weakness is that the lack of ablations,  paper does not fully isolate which gains come from the analytic transition formulation versus the refinement strategy or search procedure and prescreening.

---

> ### Author Rebuttal · Authors · 2026-03-30
>
> We gratefully acknowledge the reviewer's careful reading and constructive comments.
>
> > **W1 / Q1: Ablation gap.** How much of the final gain comes from the RL objective vs. refinement vs. prescreening?
>
> **A1: We provide a full component ablation across all 23 PMO tasks** (AUC top-10 averaged, all oracle calls including refinement are counted toward the same 10k budget).
>
> |Method|RL|Refine|Screen|Dynamic Prior|AUC-top10|
> |---|---|---|---|---|---|
> |DeFoG|-|-|-|-|11.079|
> |DeFoG|-|✓|-|-|15.251|
> |DeFoG|-|✓|✓|-|15.887|
> |Graph-GRPO|✓|-|-|-|16.901|
> |Graph-GRPO|✓|-|-|✓|17.450|
> |Graph-GRPO|✓|✓|-|-|17.950|
> |Graph-GRPO|✓|✓|-|✓|18.987|
> |Graph-GRPO|✓|✓|✓|-|18.033|
> |Graph-GRPO|✓|✓|✓|✓|**19.270**|
>
> Note: All Graph-GRPO results in the main paper (Table 4) include dynamic prior by default.
>
> **Key findings:**
>
> 1. **RL is the dominant contributor** (+5.82 over DeFoG), as it directly optimizes the task reward.
> 2. **Refinement** adds +4.17 without RL, but +1.05 to +1.54 with RL. **Prescreening** adds +0.64 without RL, +0.08 to +0.28 with RL. The diminishing gains are expected: RL already steers the model toward high-reward regions, reducing the marginal benefit of search-based components.
> 4. **Dynamic prior** contributes +0.55 to +1.24. Its effect is larger when combined with refinement, as it concentrates the prior on high-reward regions, providing better starting points for refinement.
>
> All components are complementary. We will include this table in the revision.
>
> > **Q2: Numerical stability.** What ensures the analytic transition probabilities remain stable across all time steps?
>
> **A2: Three mechanisms ensure numerical stability, combining the rate matrix formulation with implementation-level safeguards.**
>
> 1. Denominator stabilization. The denominator of the analytic rate matrix (Eq. 10)  contains $Z_t^{>0}(1-t)p_0(z_t)$, which approaches zero as $t \to 1$. We add a small constant $\epsilon = 10^{-6}$ to prevent division-by-zero.
>
> 2. Non-negative rates and exact normalization (Eq. 10, 29). The conditional rate matrix formulation [5] uses ReLU to guarantee non-negative off-diagonal rates. The staying probability is computed as $p(z_{t+\Delta t} = z_t | z_t) = 1 - \sum_{z_{t+\Delta t} \neq z_t} p(z_{t+\Delta t} | z_t)$ (Eq. 29), rather than from the diagonal rate directly, to ensure the transition distribution sums to exactly 1 despite floating-point precision.
>
> 3. Log-domain clamping. Before taking logarithms during RL training, all transition probabilities are clamped to a minimum of $10^{-8}$, preventing $\log(0)$ from floating-point underflow.
>
> We will add a detailed description of these numerical stability measures in the revision.
>
> > **Q3: Node/edge factorization.** Does this assumption limit applicability on graphs with stronger structural dependencies?
>
> **A3: This is a necessary trade-off for efficiency.**
>
> 1. Without factorization, the joint state space is $|\mathcal{X}|^N \times |\mathcal{E}|^{N^2}$, which is intractable for both training and inference. The independence assumption (Eq. 1) reduces this to $N \times |\mathcal{X}| + N^2 \times |\mathcal{E}|$.
> 2. This assumption is widely adopted by prior discrete graph generation models ([1]; DeFoG [2]; DiGress [3]; Cometh [4]). Our experimental results on both synthetic graphs (Planar, Tree) and molecular graphs suggest the practical impact on generation quality is limited.
> 3. The GraphTransformer backbone partially compensates for this assumption by capturing structural dependencies through attention over the full graph topology, making the predicted distribution $p_\theta(\cdot|z_t)$ context-aware despite the factorized transition.
>
> **References:**
>
> [1] Campbell et al., "A continuous time framework for discrete denoising models", NeurIPS 2022.
>
> [2] Qin et al., "DeFoG: Discrete flow matching for graph generation", ICML 2025.
>
> [3] Vignac et al., "DiGress: Discrete denoising diffusion for graph generation", ICLR 2023.
>
> [4] Siraudin et al., "Cometh: A continuous-time discrete-state graph diffusion model", TMLR 2025.
>
> [5] Campbell et al., "Generative flows on discrete state-spaces: Enabling multimodal flows with applications to protein co-design", ICML 2024.

---

> > ### Author Rebuttal · Reviewer_vXSd · 2026-04-04
> >
> > Thank you for the clarification. I keep my score as weak accept.

---

### Decision · Program_Chairs · 2026-04-30

**Decision:**

Accept (regular)

**Comment:**

This paper makes an interesting attempt to introduce reinforment learning into the training of graph flow models. In the rebuttal, the authors provide addtional ablations and validate the effectiveness of analytic transition over other RL methods, such as REINFORCE. The experiments are sufficient and the results are promising.

The concerns of all reviewers are fully resolved. Three reviewers give positive recommendations (1 accept and 2 weak accept). Therefore, I recommend accepting this paper.

The authors should update the additional ablation experiments and resolved the questions in the revision.